🔓 | **Open Peer Review** | Bacteriology | Research Article

# Glycoproteomic and proteomic analysis of *Burkholderia cenocepacia* reveals glycosylation events within FliF and MotB are dispensable for motility

Jessica M. Lewis,[1] Leila Jebeli,[1] Pauline M. L. Coulon,[1] Catrina E. Lay,[1] Nichollas E. Scott[1]

**ABSTRACT** Across the Burkholderia genus *O*-linked protein glycosylation is highly conserved. While the inhibition of glycosylation has been shown to be detrimental for virulence in *Burkholderia cepacia* complex species, such as *Burkholderia cenocepacia*, little is known about how specific glycosylation sites impact protein functionality. Within this study, we sought to improve our understanding of the breadth, dynamics, and requirement for glycosylation across the *B. cenocepacia O*-glycoproteome. Assessing the *B. cenocepacia* glycoproteome across different culture media using complementary glycoproteomic approaches, we increase the known glycoproteome to 141 glycoproteins. Leveraging this repertoire of glycoproteins, we quantitively assessed the glycoproteome of *B. cenocepacia* using Data-Independent Acquisition (DIA) revealing the *B. cenocepacia* glycoproteome is largely stable across conditions with most glycoproteins constitutively expressed. Examination of how the absence of glycosylation impacts the glycoproteome reveals that the protein abundance of only five glycoproteins (BCAL1086, BCAL2974, BCAL0525, BCAM0505, and BCAL0127) are altered by the loss of glycosylation. Assessing ΔfliF (ΔBCAL0525), ΔmotB (ΔBCAL0127), and ΔBCAM0505 strains, we demonstrate the loss of FliF, and to a lesser extent MotB, mirror the proteomic effects observed in the absence of glycosylation in Δ*pglL*. While both MotB and FliF are essential for motility, we find loss of glycosylation sites in MotB or FliF does not impact motility supporting these sites are dispensable for function. Combined this work broadens our understanding of the *B. cenocepacia* glycoproteome supporting that the loss of glycoproteins in the absence of glycosylation is not an indicator of the requirement for glycosylation for protein function.

**IMPORTANCE** *Burkholderia cenocepacia* is an opportunistic pathogen of concern within the Cystic Fibrosis community. Despite a greater appreciation of the unique physiology of *B. cenocepacia* gained over the last 20 years a complete understanding of the proteome and especially the O-glycoproteome, is lacking. In this study, we utilize systems biology approaches to expand the known *B. cenocepacia* glycoproteome as well as track the dynamics of glycoproteins across growth phases, culturing media and in response to the loss of glycosylation. We show that the glycoproteome of *B. cenocepacia* is largely stable across conditions and that the loss of glycosylation only impacts five glycoproteins including the motility associated proteins FliF and MotB. Examination of MotB and FliF shows, while these proteins are essential for motility, glycosylation is dispensable. Combined this work supports that *B. cenocepacia* glycosylation can be dispensable for protein function and may influence protein properties beyond stability.

**KEYWORDS** glycosylation, *Burkholderia cenocepacia*, *Burkholderia*, post-translational modifications, proteomics, PglL, glycoproteomics

Address correspondence to Nichollas E. Scott, Nichollas.scott@unimelb.edu.au.

Jessica M. Lewis, Leila Jebeli, and Pauline M. L. Coulon contributed equally to this article. Joint first author order was determined by length of time spend working on the project. This project was a true team effort involving multiple team members over the course of three years.

The authors declare no conflict of interest.

See the funding table on p. 19.

The *Burkholderia cepacia* complex (Bcc) is a group of at least 20 phenotypically similar yet genetically distinct species of increasing clinical importance (1–3). Bcc members are associated with chronic and life-threatening infections within sensitive populations, such as individuals with cystic fibrosis (CF) (1, 4), where they can establish persistent infections (4, 5). Bcc colonization of the CF lung is associated with accelerated lung function decline and reduced life expectancy (6, 7), leading to elevated rates of morbidity and mortality compared to other bacterial infections (8–11). While aggressive antibiotic regimes are utilized to control these infections (12), the intrinsic resistance of Bcc members makes the eradication of these infections challenging (13). *Burkholderia cenocepacia*, a predominant member of the Bcc, is of particular concern to the CF community as it is associated with a heightened risk of developing "Cepacia syndrome," a rapidly necrotizing form of pneumonia that can result in fulminating septicemia, which is nearly uniformly fatal (4, 14). While it is now recognized that *B. cenocepacia* infections are associated with poor clinical prognosis (15) the underlying drivers of these clinical differences are still unclear.

Over the past 20 years, significant strides have been made in our understanding of *B. cenocepacia* CF lung colonization (6, 7), yet how the repertoire of known *B. cenocepacia* virulence factors (16, 17) is coordinated under these conditions remains largely unclear. Within the CF lung, *B. cenocepacia* is observed extracellularly within luminal mucus as well as within macrophages of the mucopurulent material (5, 18). Due to growth under similar conditions, several CF-like media models established for *Pseudomonas aeruginosa*, including artificial sputum medium (ASM) (19, 20) and synthetic cystic fibrosis medium (21), have been used to probe the response of *B. cenocepacia* to extracellular growth in CF-like conditions. Indeed, multiple metabolic (22) and transcriptomic (23–25) studies have now shown that under CF-like conditions, *B. cenocepacia* reveals marked physiological alterations such as changes in motility and flagellin expression (26), as well as shifts in metabolism toward phenylacetic acid degradation (23, 27). While these studies have proven insightful for understanding transcriptional changes, whether these changes reflect proteome-level events remains unclear (28). As post-translational regulation is important for controlling virulence within *B. cenocepacia*, as noted for the iron starvation σ factor OrbS (25), understanding the proteome changes under CF-like conditions may provide valuable insights into the regulation of virulence. This is especially true for post-translational protein modifications, such as glycosylation, which may be influenced by the interplay of glycosylation machinery, sugar precursor levels, and protein substrate availability.

While once thought to be absent in bacterial systems, glycosylation is now known to be commonplace (29, 30) within multiple species including members of the *Campylobacter* (31, 32), *Neisseria* (33–37), and *Burkholderia* (38–40) genera. Within *Burkholderia* species multiple proteins are targeted for glycosylation by a conserved general *O*-linked glycosylation system encoded by two non-genetically linked components; the <u>O</u>-linked <u>G</u>lycosylation <u>C</u>luster (*ogc*), responsible for the biosynthesis of the trisaccharide β-Gal-(1,3)–α-GalNAc-(1,3)–β-GalNAc used for *O*-linked glycosylation (40); and the *O*-linked oligosaccharyltransferase PglL, required for the *en bloc* transfer of the *O*-glycan to serine residues of compatible periplasmic proteins (38, 41). Within *B. cenocepacia* the loss of general *O*-linked glycosylation leads to dramatic changes in virulence (40, 41) with similar impacts on virulence also noted in the non-CF pathogen *Burkholderia pseudomallei* (42). While these observations support the involvement of *O*-linked glycosylation in *Burkholderia* virulence the specific glycoproteins and glycosylation sites remain unclear. Recent studies have now expanded the known *B. cenocepacia* glycoproteome to more than >100 glycoproteins yet most glycoproteins have no known function (38, 39, 43). Furthermore, no information is currently available on the abundance of these glycoproteins under differing growth conditions including host-like stimuli. Thus, to improve our understanding of the *B. cenocepacia* glycoproteome and define which glycoproteins may be associated with virulence further insights into the dynamics of glycoproteins are needed.

The analysis of bacterial proteomes using liquid chromatography–mass spectrometry (LC-MS) has emerged as the quintessential approach for the quantitation of microbial proteomes and post-translational modifications (44, 45). Within bottom-up proteomics, the most widely implemented form of proteomics, proteins are first digested into peptides prior to analysis (46). Once digested, peptides can be analyzed directly or subjected to enrichment approaches to enhance the detection of specific peptides of interest, such as glycopeptides in the case of glycoproteomics (47). In glycoproteomics, a range of enrichment approaches can be used to isolate glycopeptides (47) with Zwitterionic hydrophilic interaction liquid chromatography (ZIC-HILIC) (38, 39, 41, 48–50) and Field Asymmetric waveform Ion Mobility Spectrometry (FAIMS) (43, 51, 52) being two commonly utilized approaches. These enrichment approaches exploit the hydrophilic nature of glycans, in the case of ZIC-HILIC, or the shape of glycopeptide ions, in the case of FAIMS, to allow the separation of glycosylated peptides from non-glycosylated peptides improving the depth of glycoproteomic analysis. For both enriched and total proteome analyses, the resulting samples are analyzed by LC-MS, where in peptides/glycopeptides are subjected to MS-based fragmentation, and the resulting spectra analyzed using computational tools which match the resulting spectra to a protein database. While traditionally bottom-up proteomics using data-dependent acquisition (DDA) has been used for bacterial proteome analysis (53–55) and the analysis of glycopeptides (38, 39, 41, 43, 48–52), the stochastic nature of this method leads to missing data points across samples compromising quantitative analysis, especially within low abundant proteins (56). To address these issues data-independent acquisition (DIA) has emerged as an alternative to DDA, which leads to a dramatic reduction in missing data points across studies while enabling deep proteome analysis (57, 58). Within DIA, the highly reproducible sampling across the LC gradient enables the reproducible selection and fragmentation of peptides even allowing the detection of peptides not observable on the MS1 level (59, 60), with recent improvements in DIA informatics broadening the accessibility of this approach (57, 61–63). Combined these advancements now allow DIA to provide deep bottom-up proteomic coverage and track even challenging protein classes such as microproteins (<100 amino acid in length) (64) as well as low abundant proteins (59, 60).

Within this study, we have sought to consolidate our understanding of the known *B. cenocepacia* glycoproteome and define the dynamics of glycoproteins under different growth conditions, growth phases, and in response to glycosylation inhibition. Complementing our previous studies (38, 39, 43), we analyzed the glycoproteome of *B. cenocepacia* across multiple growth conditions identifying that at least 141 glycoproteins are subjected to glycosylation within *B. cenocepacia*. Using DIA analysis, we demonstrate most glycoproteins are stably expressed across growth phases and culturing conditions yet a subset of five glycoproteins undergo dramatic reductions in protein abundance in the absence of glycosylation. Of these glycoproteins, we demonstrate the glycoproteins FliF (BCAL0525) and MotB (BCAL0127) are essential for motility yet the known glycosylation sites within these proteins appear dispensable for protein function. Overall, this study expands the known *B. cenocepacia* glycoproteome and demonstrates glycosylation sites, even within proteins that appear impacted by the loss of glycosylation, can be dispensable for protein functionality.

## MATERIALS AND METHODS

### Bacterial strains and growth conditions

Strains used in this study are listed in Table S10. Strains of *B. cenocepacia* were grown at 37°C in liquid cultures to the desired growth phase in one of three different media: Lysogeny Broth (LB), Tryptic Soy Broth (TSB) or Artificial Sputum Media (ASM). Cultures were inoculated 1/10 with 0.02 $OD_{600nm}$ of overnight LB cultures and allowed to grow for either 8 or 15 h (defined as Log and Stationary phase respectively; Fig. S1) with shaking at 185 rpm. LB and TSB media were prepared according to the manufacturer's

instructions (BD, New Jersey, USA), while the ASM was prepared according to previous reports (65–67). Individual components of the ASM, except for egg yolk emulsion (Oxiod/ Thermo Fisher Scientific) and porcine mucin (Sigma), were filter sterilized using a 0.22-µM filter. Porcine stomach mucin was sterilized by dissolving in Milli-Q water, 0.25 mg/mL, then autoclaving. After sterilization/filtering, individual components of the ASM were combined to a final concentration of 10 mg/mL mucin (Sigma-Aldrich), 1.39 mg/mL salmon sperm DNA (Sigma-Aldrich), 5 mg/mL NaCl, 2.2 mg/mL KCl, 0.22 mg/mL $CaCl_2$, 5 mg/mL Casamino acids (Bacto/Thermo Fisher Scientific, USA), 10 mg/mL Bovine Serum Albumin (Bovogen Biologicals, Australia), 0.5% egg yolk emulsion, and 0.005% diethylenetriaminepentaacetic acid (Sigma) in NaOH adjusted Milli-Q water (pH 14, Sigma-Aldrich). The pH of the final media was confirmed to be ~7 and the sterility of batches was checked by plating 100 µL of artificial sputum media onto LB plates and incubating for 24 h at 37°C. *B. cenocepacia* strains were grown to the required growth phase within the media of interest and then collected by centrifugation at 10,000 × *g* at 4°C for 10 min and washed three times with ice-cold phosphate-buffered saline (PBS) to remove media contaminates. Washed cells were then snap frozen and then stored at −80°C until processing.

When required, antibiotics were added to a final concentration of 50 µg/mL trimethoprim for *Escherichia coli* and 100 µg/mL for *B. cenocepacia*, 20 µg/mL tetracycline for *E. coli*, and 150 µg/mL for *B. cenocepacia* and 40 µg/mL kanamycin for *E. coli*. Ampicillin was used at 100 µg/mL and polymyxin B at 25 µg/mL for triparental mating to select against donor and helper *E. coli* strains as previously described (68). Induction of complemented *B. cenocepacia* strains was undertaken by the addition of L-rhamnose monohydrate (Sigma-Aldrich, final concentration 0.1%) to agar plates used for motility assays as well as liquid cultures used for western blotting and proteomic experiments. Antibiotics were purchased from Thermo Fisher Scientific while all other chemicals, unless otherwise stated, were provided by Sigma-Aldrich.

## Recombinant DNA methods

Oligonucleotides used in this study are provided in Table S12. pGPI-SceI (69) mutagenesis constructs pGPI-SceI-Δ*fliF* (BCAL0525), pGPI-SceI-ΔBCAM0505, and pGPI-SceI-Δ*motB* (BCAL0127) were generated using Gibson assembly (70) by the insertion of PCR amplified fragments into *Sma*I linearized pGPI-SceI using NEBuilder HiFi DNA master mix according to the manufacturer's instructions (New England Biolabs). The inducible pSCrhaB2-*fliF*-his$_{10}$ and pSCrhaB2-*motB*-his$_{10}$ constructs were generated using Gibson assembly (70) by inserting PCR amplified products into *Nde*I and *Xba*I linearized pSCrhaB2 (71) using NEBuilder HiFi DNA master mix. pSCrhaB2 site-directed mutagenesis was undertaken using PCR-based site replacement and *Dpn*I digestion (72). All restriction endonuclease digests, and agarose gel electrophoresis were performed using standard molecular biology techniques with chemically competent *E. coli* pir2 cells transformed using heat shock-based transformation (72). PCR amplifications were carried out using Q5 DNA polymerase (New England Biolabs) according to the manufacturer's recommendations with the addition of 2% DMSO for the amplification of *B. cenocepacia* DNA, due to its high GC content. Genomic DNA isolations were performed using genomic DNA clean-up Kits (Zymo Research), while PCR recoveries and restriction digest purifications were performed using DNA Clean & Concentrator Kits (Zymo Research). Colony and screening PCRs were performed using GoTaq DNA polymerase (Promega). Plasmids were confirmed by either Sanger sequencing undertaken at the Australian Genome Research Facility (Melbourne, Australia) or nanopore plasmid sequencing using Plasmidsaurus (SNPsaurus LLC, Eugene, OR) with a complete list of the plasmids used within this study provided in Table S11.

## Construction of unmarked Δ*fliF*, Δ*motB*, and ΔBCAM0505 deletions and the introduction of pSCrhaB2 plasmids into *B. cenocepacia*

Deletions of genes of interest were undertaken using the approach of Flannagan *et al.* for the construction of unmarked, non-polar deletions in *B. cenocepacia* (69). Plasmids were introduced into *B. cenocepacia* K56-2 (73) or *B. cenocepacia* K56-2 Δ*pglL* (55) by triparental mating with the aid of *E. coli* containing pRK2013 (74). Removal of the integrated pGPI-SceI plasmids to generate required deletions was achieved by the introduction of pDAI-SceI-SacB (69, 75) by triparental mating. Mutagenesis was confirmed using screening oligonucleotides which bind outside the region of recombination (Table S12) using GoTaq DNA polymerase-based PCR screening supplemented with 10% DMSO. Complementation (pSCrhaB2) plasmids were introduced into strains of interest by triparental mating with the aid of *E. coli* containing pRK2013 (74) with confirmation of complementation with the correct plasmid undertaken using PCR-based screening and Sanger sequencing.

## Preparation of proteomic samples

Frozen whole-cell samples were prepared for analysis using the in-StageTip preparation approach as previously described (76). Cells were resuspended in 4% sodium deoxycholate (SDC), 100 mM Tris pH 8.0 and boiled at 95°C with shaking (2,000 rpm) for 10 min to solubilize the proteome. Samples were allowed to cool for 10 min and then boiled for a further 10 min (95°C, 2,000 rpm) before the protein concentrations were determined by bicinchoninic acid assays (Thermo Fisher Scientific). For Zwitterionic Hydrophilic Interaction Liquid Chromatography (ZIC-HILIC) glycoproteomic enrichments, 1 mg of protein from four biological replicates of each growth media at stationary phase were prepared for glycopeptide enrichment. For FAIMS-based glycopeptide enrichment and quantitative proteomic comparisons, 100 µg of protein from four biological replicates of each condition at both log and stationary phases were prepared. Samples were reduced/alkylated with the addition of Tris-2-carboxyethyl phosphine hydrochloride and Chloroacetamide (final concentration 10 and 40 mM, respectively), and samples were incubated in the dark for 1 h at 45°C. Following reduction/alkylation samples were digested overnight with Trypsin (1/50 wt/wt Solu-trypsin, Sigma) at 37°C with shaking at 1,000 rpm.

Samples for ZIC-HILIC glycoproteomic enrichment were acidified to a final concentration of 0.1% TFA to precipitate the SDC with the SDC removed by centrifugation at 10,000 × *g* for 10 min (77, 78). Clarified peptide solutions were loaded on conditioned 50 mg tC18 Sep-Pak columns (Waters Corporation, Milford, MA, USA) and peptides were desalted according to the manufacturer's instructions. tC18 Sep-Pak columns were conditioned with 10bed volumes of Buffer B (0.1% formic acid, 80% acetonitrile), then equilibrated with 10 bed volumes of Buffer A* (0.1% TFA, 2% acetonitrile) before use. Samples were loaded onto equilibrated columns then columns washed with at least 10 bed volumes of Buffer A* before bound peptides were eluted with Buffer B. Eluted peptides were dried by vacuum centrifugation at room temperature and stored at −20°C. Quantitative proteomic samples were mixed with 1.25 volumes of isopropanol to prevent the precipitation of SDC (79) and samples were acidified with TFA to a final concentration of 1% TFA. In-house-made SDB-RPS StageTips were prepared according to previously described protocols (76, 80, 81) with five frits of SDB-RPS excised using a blunt 16-gauge Hamilton needle and loaded into 200 µL tips. SDB-RPS StageTips were placed in a Spin96 tip holder (80) to enable batch-based spinning of samples and tips conditioned with 100% acetonitrile; followed by 30% methanol, 1% TFA followed by 90% isopropanol, 1% TFA with each wash spun through the column at 1,000 × *g* for 3 min. Acidified isopropanol/peptide mixtures were loaded onto the SDB-RPS columns and spun through tips before being washed with 90% isopropanol, 1% TFA followed by 1% TFA in Milli-Q water. Peptide samples were eluted with 80% acetonitrile, 5% ammonium hydroxide, and dried by vacuum centrifugation at room temperature and stored at −20°C.

## ZIC-HILIC enrichment of glycopeptides

ZIC-HILIC enrichment was performed as previously described with minor modifications (82). ZIC-HILIC Stage-tips (81) were created by packing 0.5 cm of 10 µm ZIC-HILIC resin (Millipore/Sigma) into 200 µL tips containing a frit of C8 Empore (Sigma) material. Prior to use, the columns were washed with Milli-Q water, followed by 95% acetonitrile, and then equilibrated with 80% acetonitrile, 1% TFA. Digested proteome samples were resuspended in 80% acetonitrile, 1% TFA. Samples were adjusted to a concentration of 5 µg/µL then loaded onto equilibrated ZIC-HILIC columns. ZIC-HILIC columns were washed with 20 bed volumes of 80% acetonitrile, 1% TFA to remove non-glycosylated peptides and bound peptides eluted with 10 bed volumes of Milli-Q water. Eluted peptides were dried by vacuum centrifugation at room temperature and stored at −20°C.

## LC-MS analysis of ZIC-HILIC based glycopeptide enrichments

ZIC-HILIC enriched samples were re-suspended in Buffer A* and separated using a two-column chromatography set-up composed of a PepMap100 C18 20 mm × 75 µm trap and a PepMap C18 500 mm × 75 µm analytical column (Thermo Fisher Scientific). Samples were concentrated onto the trap column at 5 µL/min for 5 min with Buffer A (0.1% formic acid, 2% DMSO) and then infused into an Orbitrap Fusion Lumos Tribrid Mass Spectrometer (Thermo Fisher Scientific) at 300 nL/min via the analytical column using a Dionex Ultimate 3000 UPLC (Thermo Fisher Scientific). 185-min analytical runs were undertaken by altering the buffer composition from 2% Buffer B (0.1% formic acid, 77.9% acetonitrile, 2% DMSO) to 28% B over 155 min, then from 22% B to 45% B over 10 min, then from 45% B to 80% B over 2 min. The composition was held at 80% B for 3 min, and then dropped to 2% B over 5 min before being held at 2% B for another 10 min. The Lumos Mass Spectrometer was operated in a data-dependent mode automatically switching between the acquisition of a single Orbitrap MS scan (maximal injection time of 50 ms, an Automated Gain Control (AGC) set to a maximum of $1*10^6$ ions and a resolution of 120k) every 3 s and Orbitrap MS/MS HCD scans of precursors (NCE 28% with 5% Stepping, maximal injection time of 60 ms, an AGC set to a maximum of $1*10^5$ ions and a resolution of 15k). Each biological replicate was analyzed two times using two different MS1 mass ranges of 350–1,800 *m/z* and 500–2,000 *m/z* to improve glycoproteome coverage. MS/MS Scans containing HexNAc-associated oxonium ions (204.0867; 138.0545 or 366.1396 *m/z*) triggered three additional product-dependent MS/MS scans (83) of potential glycopeptides; an Orbitrap EThcD scan (NCE 15%, maximal injection time of 250 ms, AGC set to a maximum of $2*10^5$ ions with a resolution of 30k and using the extended mass range setting to improve the detection of high mass glycopeptide fragment ions (84); an ion trap CID scan (NCE 35%, maximal injection time of 40 ms, an AGC set to a maximum of $5*10^4$ ions) and a stepped collision energy HCD scan (using NCE 35% with 8% Stepping, maximal injection time of 150 ms, an AGC set to a maximum of $2*10^5$ ions and a resolution of 30k).

## LC-MS analysis of FAIMS-based glycopeptide enrichments

SDB-RPS cleaned-up samples were re-suspended in Buffer A* and separated using a two-column chromatography set-up composed of a PepMap100 C18 20 mm × 75 µm trap and a PepMap C18 500 mm × 75 µm analytical column (Thermo Fisher Scientific) coupled to a Orbitrap Fusion Lumos Tribrid Mass Spectrometer equipped with a FAIMS Pro interface (Thermo Fisher Scientific). 145-min gradients were run for each sample altering the buffer composition from 2% Buffer B to 28% B over 126 min, then from 28% B to 40% B over 9 min, then from 40% B to 80% B over 3 min, the composition was held at 80% B for 2 min, and then dropped to 2% B over 2 min and held at 2% B for another 3 min. The Lumos Mass Spectrometer was operated in a stepped FAIMS data-dependent mode at three different FAIMS CVs −25, −45 and −65 as previously described (43). For each FAIMS CV a single Orbitrap MS scan (450–2,000 *m/z*, maximal injection time of 50 ms, an AGC of maximum of $4*10^5$ ions and a resolution of 60k)

was acquired every 1.2 s, followed by Orbitrap MS/MS HCD scans of precursors (NCE 30%, maximal injection time of 80 ms, an AGC set to a maximum of $1.25*10^5$ ions and a resolution of 30k). HCD scans containing HexNAc-associated oxonium ions (204.0867, 138.0545, and 366.1396 *m/z*) triggered three additional product-dependent MS/MS scans (83) of potential glycopeptides; an Orbitrap EThcD scan (NCE 15%, maximal injection time of 250 ms, AGC set to a maximum of $2*10^5$ ions with a resolution of 30k using the extended mass range setting to improve the detection of high mass glycopeptide fragment ions (84); an ion trap CID scan (NCE 35%, maximal injection time of 40 ms, an AGC set to a maximum of $5*10^4$ ions) and a stepped collision energy HCD scan (using NCE 35% with 5% Stepping, maximal injection time of 250 ms, an AGC set to a maximum of $2*10^5$ ions and a resolution of 30k).

## LC-MS analysis of DIA samples

SDB-RPS cleaned-up samples were resuspended in Buffer A* and separated using a two-column chromatography set-up composed of a PepMap100 C18 20 mm × 75 µm trap and a PepMap C18 500 mm × 75 µm analytical column (Thermo Fisher Scientific). Samples were concentrated onto the trap column at 5 µL/min for 5 min with Buffer A and then infused into an Orbitrap Eclipse (Thermo Fisher Scientific) at 300 nL/min via the analytical column using a Dionex Ultimate 3000 UPLC (Thermo Fisher Scientific). 95-min analytical runs were undertaken by altering the buffer composition from 3% Buffer B to 25% B over 82 min, then from 25% B to 40% B over 4 min, and then from 40% B to 80% B over 1 min. The composition was held at 80% B for 3 min, and then dropped to 3% B over 1 min before being held at 3% B for another 4 min. Data were collected in a data-independent manner with a single MS1 event (120k resolution, AGC $1*10^6$, 350–1,400 *m/z*) and 50 MS2 scans (NCE 30%, 30k resolution, AGC $1*10^6$, 200–2,000 *m/z* and a maximal injection time of 55 ms) of a width of 13.7 *m/z* collected over the mass range of 360 to 1,033.5 *m/z*.

## Glycopeptide identifications

ZIC-HILIC and FAIMS enrichments were analyzed using MSfragger-Glyco within MSFragger (versions 17.1) (85–87). Samples were searched with a Tryptic specificity allowing a maximum of two missed cleavage events and Carbamidomethyl set as a fixed modification of Cysteine while oxidation of Methionine allowed as a variable modification. The *Burkholderia* glycans HexHexNAc$_2$ (elemental composition: $C_{22}O_{15}H_{36}N_2$, mass: 568.2115 Da) and Suc-HexHexNAc$_2$ (elemental composition: $C_{26}O_{18}H_{40}N_2$, mass: 668.2276 Da) were included as variable modifications at Serine in line with the strong preference for PglL glycosylation at Serine residues (38). The glycan fragments of ions were defined as 204.0866, 186.0760, 168.0655, 366.1395, 144.0656, 138.0550, 466.1555, and 407.1594. For HCD searches, y, b, Y, y~/b~ (y and b ion with the addition of HexNAc) were allowed, while for EThcD searches, the ions c, z, y and y ~ ions were allowed. A maximum mass precursor tolerance of 20 ppm was allowed at both the MS1 and MS2 levels. Samples were searched against the *B. cenocepacia* reference proteome J2315 (UniProt accession: UP000001035, 6,993 proteins, downloaded 24 July 2020) (88) to enable the mapping of proteins to consensus gene accessions. HCD and EThcD searches were performed separately on each data set with the resulting "psm.tsv" combined using R with the best identification for each scan event retained and only glycopeptides with an MSfragger Hyperscore >20 used for data visualization undertaken using ggplot2 (89) using R and RStudio. To ensure high data quality, assigned glycopeptides were manually assessed and the HCD and EThcD spectra assigned to each unique glycopeptide annotated with the Interactive Peptide Spectral Annotator (90) (http://www.interactivepeptidespectralannotator.com/PeptideAnnotator.html) with annotated glycopeptides provided within Supplementary Data 1 deposited within Figshare (https://doi.org/10.6084/m9.figshare.25492774.v1). To assess the uniqueness of identified glycoproteins within this study we curated a list of all previously known *B. cenocepacia* glycoproteins (38, 39, 41, 43). Unique glycopeptides identified using the

glycopeptide identification tool Byonic [Protein Metrics Inc (91)]. yet not previously manually confirmed from studies (39, 43) were annotated to ensure correctness resulting in a total of 129 unique glycoproteins across all studies (38, 39, 41, 43) (Table S2) with unique glycopeptides from these previous Byonic searches provided within Supplementary Data 2 deposited within Figshare (https://doi.org/10.6084/m9.figshare.25492774.v1).

### Quantitative proteomics analysis using Spectronaut

DIA data sets were searched using DIA-library free analysis within Spectronaut (version 15.5, 17.6, or 18.5). Data files were searched against the *B. cenocepacia* K56-2 proteome (UniProt accession: UP000011196, 7467 proteins, downloaded 13 October 2020) (92) and *B. cenocepacia* strain J2315 (UniProt accession: UP000001035) (88), effectively merging these proteomes enabling the matching of proteins to both J2315 and K56-2 accessions. Oxidation of Methionine was allowed as a variable modification, Carbamidomethyl set as a fixed modification of Cysteine and the protease specificity set to Trypsin. Protein quantitation was undertaken using MaxLFQ (93) based analysis. The precursor PEP was altered to 0.01 from the default 0.2 to improve quantitative accuracy with all single peptide protein matches excluded. Statistical analysis was undertaken using Perseus (94) with missing values imputed based on the total observed protein intensities with a range of 0.3 σ and a downshift of 2.5 σ. Biological replicates were grouped together and either ANOVA analysis was used to compare changes across all groups or student *t* tests were used to compare individual groups with a minimum fold change of ± considered for further analysis. Multiple hypothesis correction was undertaken using a permutation-based FDR approach allowing an FDR of 5%. Enrichment analysis using Fisher exact tests was undertaken in Perseus using gene ontology (GO) terms obtained from UniProt (*B. cenocepacia* strain J2315 proteome: UP000001035). Data visualization was undertaken using ggplot2 (89) within RStudio.

### DDA confirmation of independent ΔfliF, ΔmotB, and ΔBCAM0505 B. cenocepacia mutants

In-StageTip prepared proteome digests as above were re-suspended in Buffer A* and separated using a two-column chromatography setup composed of a PepMap100 C18 20 mm × 75 µm trap and a PepMap C18 500 mm × 75 µm analytical column (Thermo Fisher Scientific). Samples were concentrated onto the trap column at 5 µL/min for 5 min with Buffer A and then infused into either an Orbitrap Q-Exactive plus, an Orbitrap Eclipse or Orbitrap Fusion Lumos equipped with a FAIMS Pro interface (Thermo Fisher Scientific) at 300 nL/min via the analytical column using Dionex Ultimate 3000 UPLCs (Thermo Fisher Scientific). 125-min analytical runs were undertaken by altering the buffer composition from 2% Buffer B to 22% B over 95 min, then from 22% B to 40% B over 10 min, and then from 40% B to 80% B over 5 min. The composition was held at 80% B for 5 min, and then dropped to 2% B over 2 min before being held at 2% B for another 8 min. The Mass Spectrometers were operated in a data-dependent mode automatically switching between the acquisition of a single Orbitrap MS scan and up to 15 Orbitrap MS/MS HCD scans of precursors on the Orbitrap Q-Exactive plus (Stepped NCE of 28%, 32%, and 38%, a maximal injection time of 100 ms, an AGC set to a maximum of $2*10^5$ ions and a resolution of 35k) or up to 3 s of MS/MS HCD scans of precursors on the Orbitrap Lumos and Eclipse (Stepped NCE of 25%, 30%, and 40%, a maximal injection time of 55 ms, an AGC set to a maximum of $2*10^5$ ions and a resolution of 30k).

### DDA proteomic data analysis

Identification and LFQ analysis to confirm *B. cenocepacia* mutants was accomplished using MaxQuant (v1.6.17.0) (95) using the *B. cenocepacia* K56-2 proteome (UniProt accession: UP000011196) (92) and *B. cenocepacia* strain J2315 (UniProt accession: UP000001035) (88) allowing for oxidation on Methionine, Carbamidomethyl set as a fixed modification on Cysteine and the protease specificity set to Trypsin. Prior to MaxQuant

analysis data sets acquired on the Fusion Lumos were separated into individual FAIMS fractions using the FAIMS MzXML Generator (96). The LFQ and "Match Between Run" options were enabled to allow comparisons between samples. The resulting data files were processed using Perseus (v1.4.0.6) (94) using Student's $t$ tests to compare proteomic changes between independent mutants with data visualization undertaken using ggplot2 (89) within RStudio.

### Western blotting

Cells grown overnight with or without L-rhamnose (0.1%; stationary phase, $OD_{600nm}$ 1.5) were resuspended in 1× SDS-PAGE sample buffer, boiled for 10 min and then run on a NuPAGE 4–12% Bis-Tris gradient polyacrylamide gel in MES buffer (Thermo Fisher Scientific). Proteins were transferred to a nitrocellulose membrane using the iBlot 2 Western transfer system (Thermo Fisher Scientific) and blocked for 1 h in 5% skim milk in TBS-T (Tris-buffered saline with 0.1% Tween-20). Nitrocellulose membranes were probed with a monoclonal mouse αHistidine tag primary antibody (1:1,000; AD1.1.10, Bio-Rad) overnight at 4°C, washed with TBS-T, then incubated with horseradish peroxidase (HRP) conjugated αmouse antibody (1:5,000; NEF82200-1EA, PerkinElmer) for 1 h at room temperature. Proteins were detected using the Clarity Western ECL Substrate (Bio-Rad), and images were obtained using an Amersham imager 600 or 800 (GE Healthcare Life Sciences). Membranes were Stripped with Restore Plus Stripping Buffer for 10 min before being re-blocked in 5% skim milk in TBS-T and probed with α-*E. coli* RNA polymerase primary antibody (1:5,000; 4RA2, Neoclone), followed by HRP conjugated αmouse and imaged as above. All antibodies were diluted in TBS-T with 1% bovine serum albumin (BSA; Sigma-Aldrich). Raw uncropped and protein marker-associated Western blotting images are provided in Fig. S24.

### Swimming motility assays

Motility assays were performed with 0.3% L-agar as previously described (26) with minor modifications. Briefly, 20% sterile L-rhamnose or an equivalent volume of sterile water was added to molten 0.3% L-agar media for a final concentration of 0.1% rhamnose, 20 mL aliquoted into 9 mm Petri dishes, and plates dried briefly once set. The dried plates were inoculated with 1 µL of stationary phase cultures standardized to an $OD_{600nm}$ of 1.0 (paired inoculation of plates with and without rhamnose) and plates were incubated inside sealed bags at 37°C for 24 h. Following incubation, the diameter of turbidity was measured, and plates were imaged.

## RESULTS

### Analysis of the *B. cenocepacia* O-glycoproteome across different growth media reveals previously unrecognized glycoproteins

While our previous work using multiple proteases has revealed 98 proteins are glycosylated undergrowth in LB within *B. cenocepacia* K56-2 (38), the true size of the *B. cenocepacia* glycoproteome remains unknown. To improve our understanding of this glycoproteome, we reasoned that growth under varying culturing conditions may enable the identification of novel glycoproteins. To assess this we examined the glycoproteome of *B. cenocepacia* K56-2 (97) under growth in two common culturing media (LB and TSB) and the CF sputum-like medium ASM. As our recent work has shown improvements in the coverage of bacterial glycoproteomes using complementary enrichment approaches (43), we utilized both ZIC-HILIC (50) and FAIMS (43) based glycopeptide enrichments. Glycopeptides were identified against the J2315 proteome with our previous work highlighting the high sequence identity within the surrounding amino acids of glycosylation sites within *B. cenocepacia* (38); however, this approach is insensitive to the detection of glycopeptides unique to K56-2. Analysis of the glycopeptides observed across these growth conditions demonstrated most glycopeptides were shared across conditions yet revealed unique glycopeptides observable in each medium [Fig. 1A; Table

S1; Supplementary Data 1 (https://doi.org/10.6084/m9.figshare.25492774.v1)]. A total of 74 distinct glycopeptides were identified corresponding to 54 unique glycoproteins of which 35 and 32 glycoproteins were observed in ZIC-HILIC and FAIMS enrichments, respectively (Fig. 1B). Consistent with previous analysis ZIC-HILIC and FAIMS glycopeptide enrichments provide access to complementary glycopeptide subsets (43) (Fig. S2A through D). To assess the uniqueness of the glycoproteins identified across multiple growth conditions, we manually curated a list of all known *B. cenocepacia* glycoproteins (38, 39, 41, 43) [129 glycoproteins in total, Table S2 and Supplementary Data 2 (https://doi.org/10.6084/m9.figshare.25492774.v1)] revealing 12 of the 54 glycoproteins identified within this study correspond to previously unrecognized *B. cenocepacia* glycoproteins (Fig. 1C; Fig. S3). These novel glycoproteins, such as the Lipase chaperone (BCAM0950, Fig. 1D), raise the number of known glycoproteins to 141 with 50% of glycoproteins defined as uncharacterised or putative proteins and >40% lacking any GO terms (Fig. S4; Table S2).

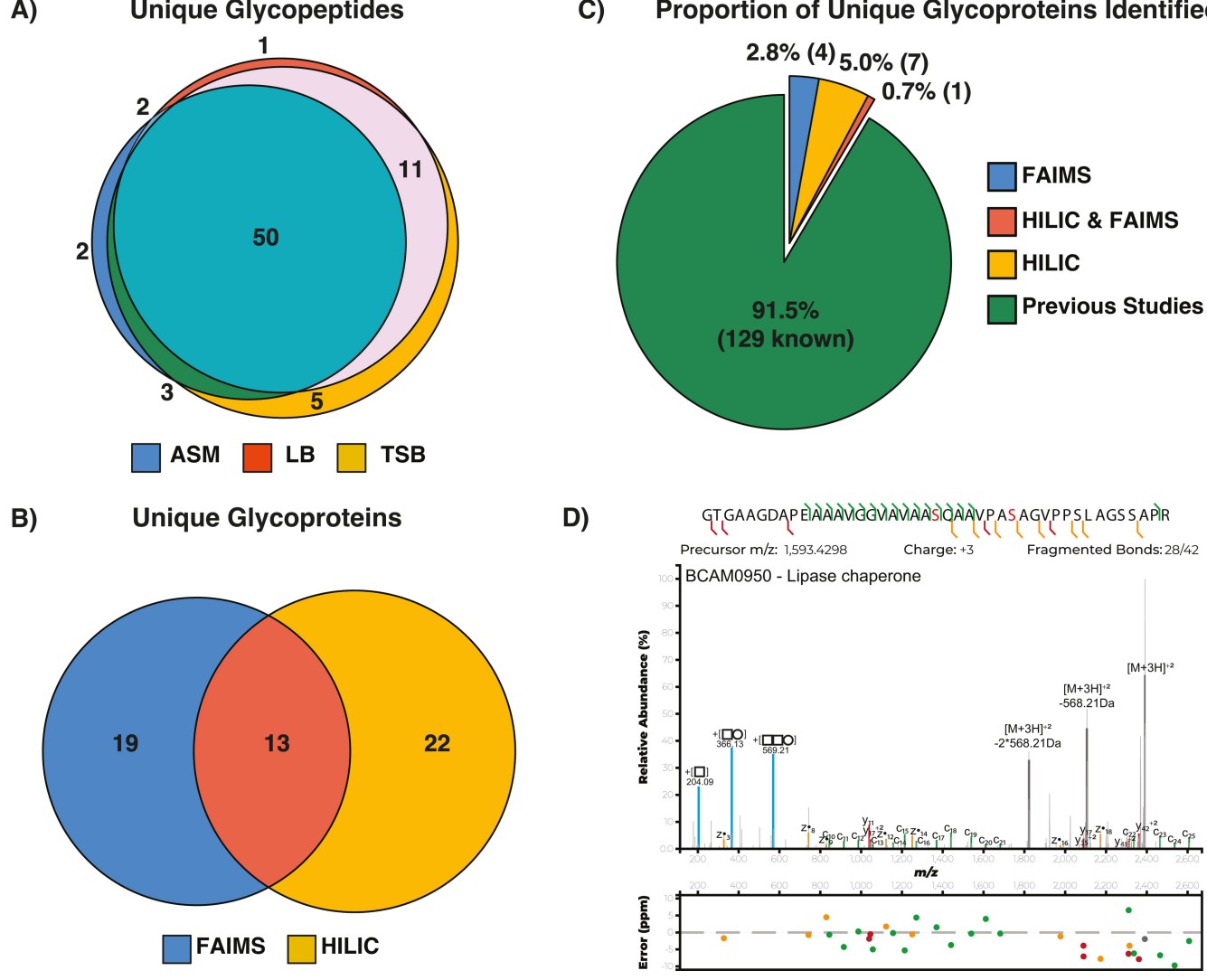

**FIG 1** Expanding the glycoproteome of *B. cenocepacia* K56-2. (A) Glycopeptide analysis reveals the majority of *B. cenocepacia* K56-2 glycopeptides appear detectable across multiple culturing conditions. (B) Venn diagram of the overlap within the identified glycoproteins across different growth conditions supports FAIMS and HILIC enrichments and provide access to complementary regions of the glycoproteome. (C) Pie chart of identified glycoproteins reveals a total of 141 glycoproteins have now been identified across *B. cenocepacia* K56-2 and J2315. (D) EThcD MS2 of the novel glycoprotein BCAM0950 reveals two glycosylation sites within the glycopeptide [39]GTGAAGDAPEAAAVGGVAVAASQAAVPASAGVPPSLAGSSAPR[81] with glycan-associated ions (204.09, 366.13, and 569.21 *m/z*, highlighted in blue) denoted according to the Symbol Nomenclature for Glycans.

## The glycoproteome is largely stable across growth phase and culturing conditions

While the identification of additional novel glycoproteins expands the known *B. cenocepacia* glycoproteome, the high degree of overlap between the glycoproteome observed under differing media and growth phases supports that a significant proportion of glycoproteins may be expressed under a range of conditions. To explore the stability of glycoprotein abundance in a robust manner, we utilized a DIA-based approach to track the known glycoproteome of *B. cenocepacia* at log and stationary phases across LB, TSB, and ASM culturing media. DIA enabled the identification of 3,923 proteins across all replicates, revealing clear clustering based on growth phases as well as culturing conditions (Fig. S5A and B; Table S3). Within the observable proteome, 106 of the 141 known glycoproteins were identified (Fig. 2A), with >90% of these glycoproteins observed in each replicate (Fig. S6A and B; Table S3). Comparing growth phases between each culturing media reveals less than 30% of the known glycoproteome appears altered in response to bacterial growth phases, while the majority of the glycoproteome appears stable across growth (Fig. 2B through E). These dynamics mirror the total proteome (Fig. S7) and support that most glycoproteins are constitutively expressed being observable across all conditions examined. Consistent with only modest alterations in glycoprotein levels across conditions, examination of the abundance of PglL (BCAL0960) and the

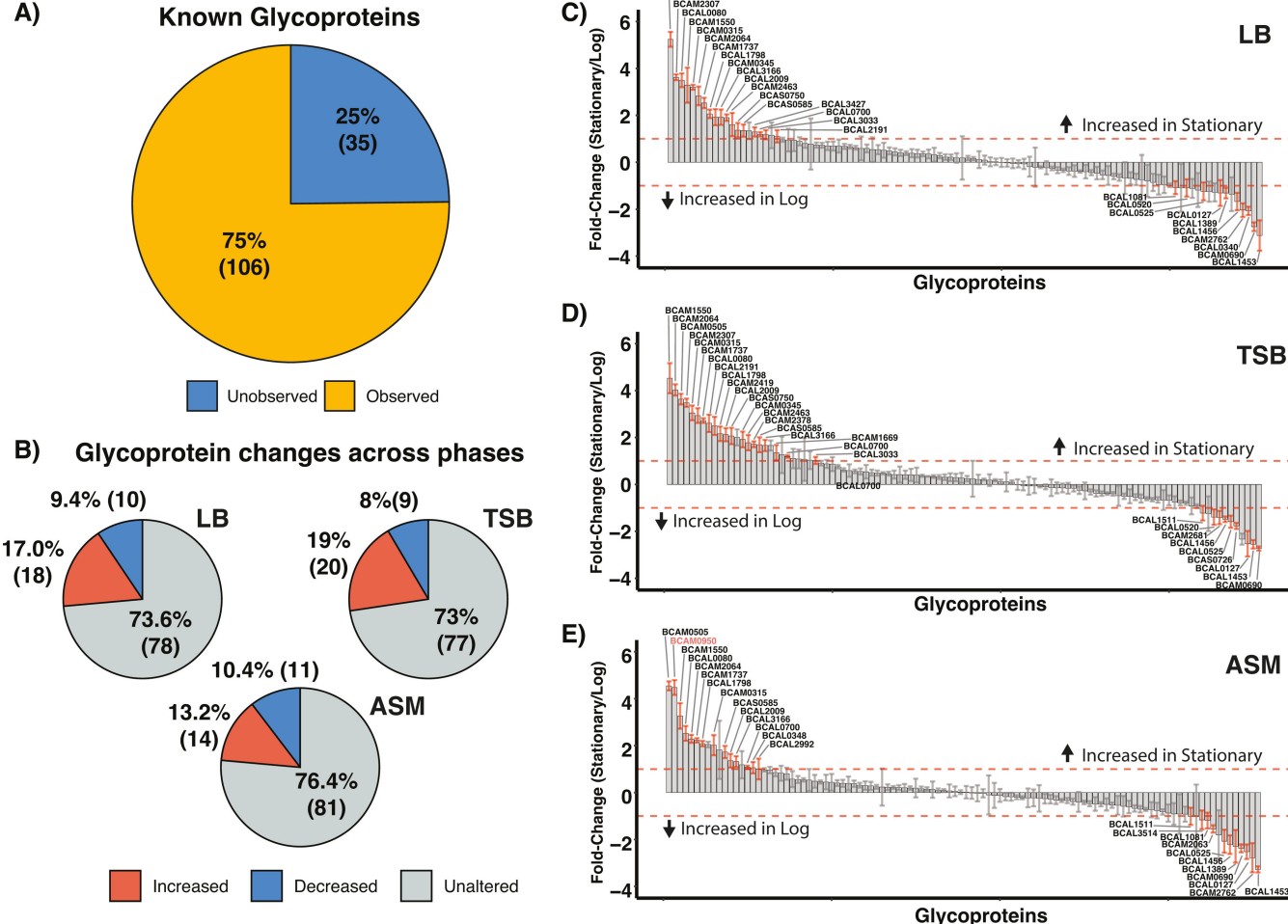

**FIG 2** DIA analysis of the *B. cenocepacia* K56-2 glycoproteome. (A) DIA analysis allows the assessment of >100 known glycoproteins across different media and growth phases. (B) Quantitative analysis of identified glycoproteins across culture media reveals most glycoproteins are unaltered across growth phases based on ±1 fold (log$_2$) and a −log$_{10}$(P value) > 2. (C–E) Rank order plots of glycoprotein levels observed across the log and stationary growth phases demonstrate only a subset of glycoproteins, denoted with red error bars, appear to be altered in abundance across growth phases.

observed members of the Ogc (BCAL3115-BCAL3117) (40) reveals constant protein levels between conditions and only modest differences ($<1$ $\log_2$) in the abundance of these proteins between growth phases (Fig. S8) supporting that glycosylation machinery abundance remains stable across growth phases and culturing conditions. While these observations support that the abundance of glycosylation substrates is mostly constant across growth phases, there were some notable exceptions such as the Lipase chaperone BCAM0950 that appeared elevated across stationary growth in ASM media (highlighted in red, Fig. 2C through E). Thus, these findings support that the majority of the glycoproteome and the associated glycosylation machinery do not appear to be impacted by changes in growth conditions or growth phases.

## The loss of glycosylation within *B. cenocepacia* K56-2 impacts only a subset of glycoproteins

Previous work has demonstrated that the loss of glycosylation results in reduced protein levels of two known glycoproteins within *B. cenocepacia* K56-2 (BCAL1086 and BCAL2974) (55). As this initial analysis only considered 21 glycoproteins known at the time, we sought to apply our DIA approach to improve the characterization of changes in the glycoproteome in the absence of glycosylation across the now known 141 glycoproteins. Examining the proteome of *B. cenocepacia* K56-2 wild type (WT), Δ*pglL* and complemented Δ*pglL* [Δ*pglL*Δ*amrAB::native-promotor-pglL-his* (55)] across differing culturing media (LB, TSB, and ASM) at stationary phase resulted in the identification of 3,783 proteins including 110 glycoproteins (Table S4; Fig. S9 and 10). Consistent with our previous work (55), we observed dramatic proteome changes between glycosylation null and glycosylation competent strains which are highly correlative across all three media conditions examined (Fig. S11). Analysis of the observable glycoproteome reveals only a subset of glycoproteins are impacted by the loss of glycosylation across all culturing media (Fig. 3A) in addition to the expected loss of AmrAB (BCAL1674-5) within the complement due to the integration of *pglL*-his into the *amrAB* locus (68, 98) (Fig. 3A). Across the glycoproteins impacted by loss of glycosylation multiple proteins appeared to undergo reductions in protein abundance, yet surprisingly two proteins, BCAL1917 and BCAL1093, also appeared to increase in abundance (Fig. 3A). Analysis of the peptides utilized for protein quantitation for BCAL1917 and BCAL1093 reveals these increases were driven by changes in peptide abundances of unmodified forms of known glycopeptide supporting this increase in abundance is an artifact resulting from the observation of peptides within Δ*pglL* that are glycosylated within WT and the *pglL*-complement strains (Fig. S12). Focusing on glycoproteins reduced in the absence of glycosylation across multiple growth media revealed the loss of five glycoproteins BCAL1086, BCAL2974, FliF (BCAL0525), BCAM0505, and MotB (BCAL0127) (Fig. 3B). Examination of the peptides quantified for these glycoproteins supports the reduction of these proteins across all growth conditions and the restoration of these proteins upon complementation (Fig. 3C). Thus, these findings support that the abundance of most glycoproteins is unaffected by the loss of glycosylation, with only five glycoproteins undergoing dramatic reductions in abundance in the absence of glycosylation.

## Loss of MotB and FliF mirrors the proteomic impacts observed in *B. cenocepacia* K56-2 Δ*pglL*

As our previous studies have demonstrated that the disruption of *BCAL1086* or *BCAL2974* does not drive the proteome changes observed in Δ*pglL* (55), we focused on the glycoproteins MotB, FliF, and BCAM0505. To assess if the loss of *motB*, *fliF*, and BCAM0505 contributes to the proteomic impact seen in the absence of glycosylation, we generated Δ*motB*, Δ*fliF*, and ΔBCAM0505 strains, and using DDA proteomic analysis confirmed the loss of these three proteins within mutants (Fig. S13 to S15; Table S5). Additional DIA analysis of Δ*motB*, Δ*fliF*, and ΔBCAM0505 compared to the WT and Δ*pglL* strains at stationary phase in LB allowed the identification of 3,934 proteins with individual biological groups demonstrating clustering by Pearson correlation and PCA analysis

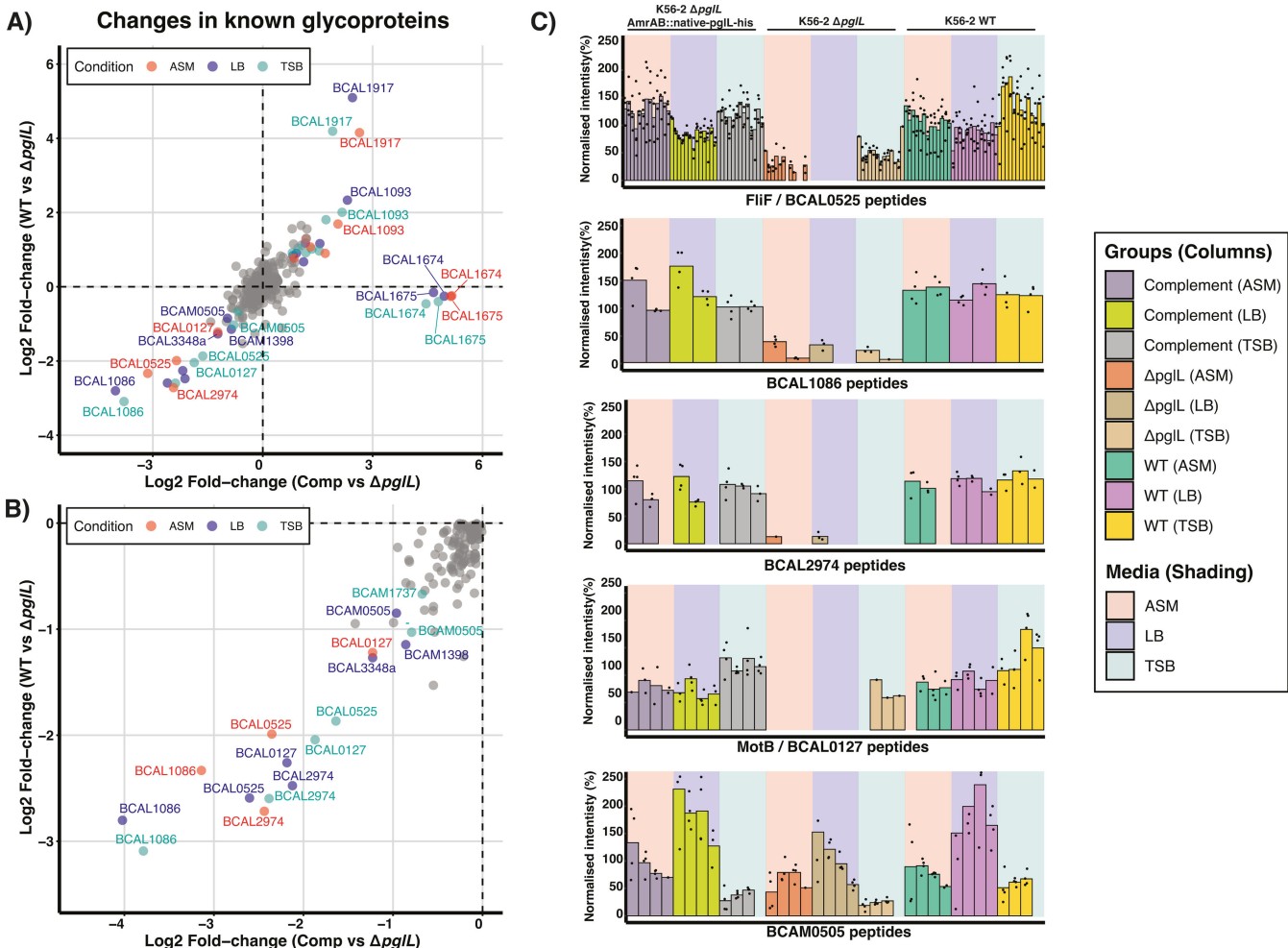

**FIG 3** Impact of the loss of glycosylation on the *B. cenocepacia* K56-2 glycoproteome. (A and B) 2D scatter plots comparing glycoprotein levels observed within WT versus Δ*pglL* and Δ*pglL amrAB::native-pglL-his* versus Δ*pglL*. Glycoproteins observed to undergo alterations are color-coded according to the growth condition if the observed alteration has a fold change greater than ±1 fold (log$_2$) and a −log$_{10}$(*P* value) > 2. (C) Peptide-centric analysis of glycoproteins FliF, BCAL1086, BCAL2974, MotB, and BCAM0505 observed to undergo alterations in response to the loss of glycosylation reveals a marked decrease in peptide intensities within these proteins which is restored by the re-introduction of glycosylation within the complement Δ*pglL amrAB::native-pglL-his*.

(Fig. S16 and 17; Table S6). Across strains, we observed that the loss of *motB*, *fliF*, and BCAM0505 leads to observable proteome alterations compared to WT [310, 117, and 46 protein alterations with a fold change of ±1 fold (log$_2$) and a −log$_{10}$(*P* value) > 2.0 respectively] with many of these alterations observed within proteins also impacted by the loss of *pglL* (Fig. 4A; Fig. S18). Enrichment analysis of the proteins altered within Δ*motB* and Δ*fliF* highlights the high degree of overlap within proteome changes observed within Δ*pglL* (Fig. 4B and C; Table S7). Additionally, enrichment analysis of GO terms associated with protein alterations within Δ*fliF* supports changes in chemotaxis and flagella-associated motility proteins (Fig. 4D; Table S7) consistent with alterations previously observed within Δ*pglL* (55). Interestingly, enrichment analysis of alterations observed within Δ*motB* did reveal functional enrichments, yet these were unique to those observed with Δ*pglL* (Fig. S19A and B; Table S7). In contrast, no enrichment in GO terms or proteins observed altered within Δ*pglL* were observed with ΔBCAM0505. Examination of the individual proteins associated with the GO terms Chemotaxis [GO:0006935] and Bacterial-type flagellum-dependent cell motility [GO:0071973] supports similar changes between Δ*pglL* and Δ*fliF* compared to WT yet that the overlap in altered proteins is only partial (Fig. S19C and D). Combined, these

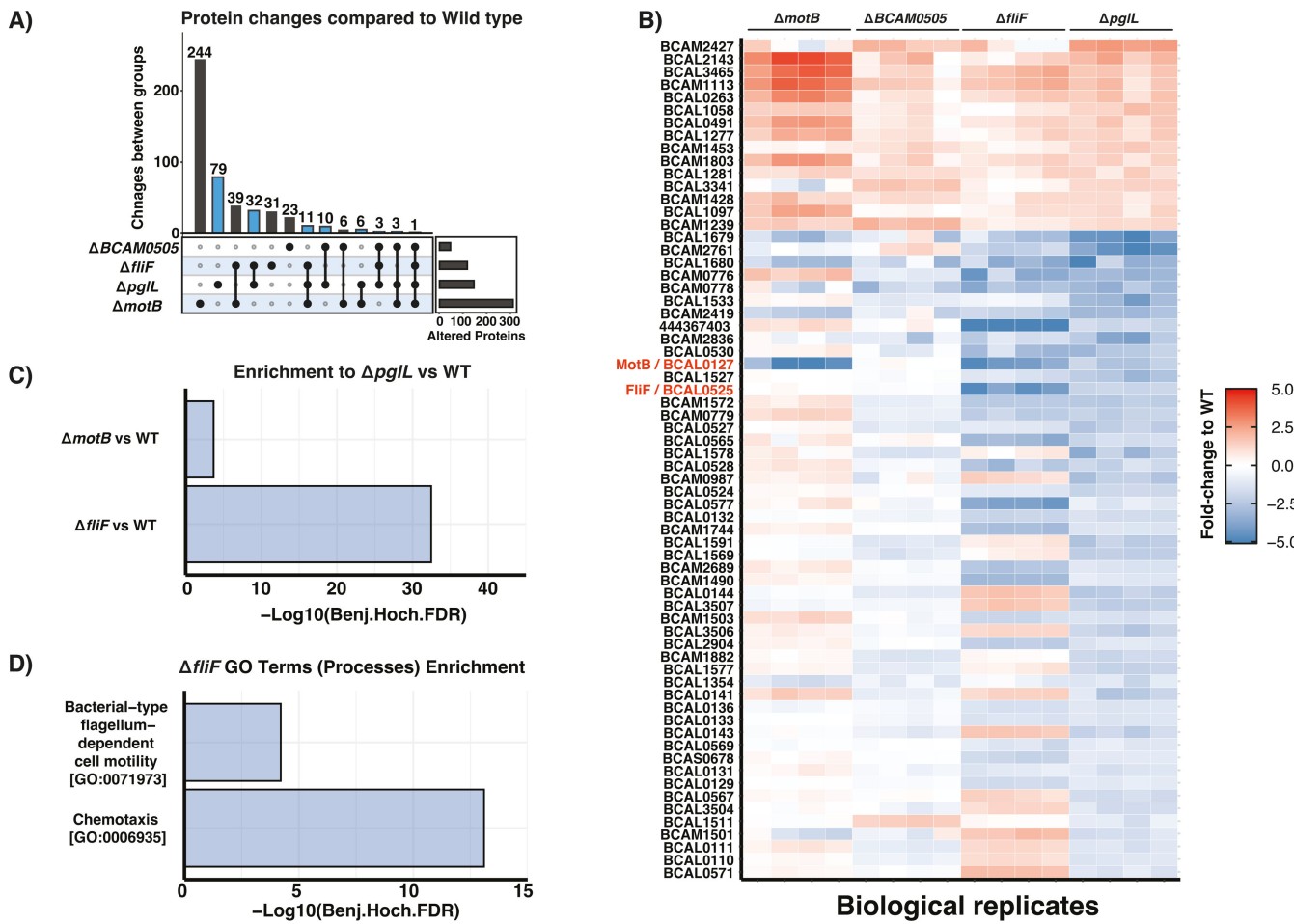

**FIG 4** Proteome analysis of ΔmotB, ΔfliF, and ΔBCAM0505 compared to WT and ΔpglL. Proteome analysis of the strains lacking the glycoproteins MotB (BCAL0127), FliF (BCAL0525), and BCAM0505 compared to WT and ΔpglL supports the loss of MotB and/or FliF mirrors proteomic impacts observed within ΔpglL. (A) Upset analysis of the proteins observed altered between strains based on ±1 fold (log₂) and a −log₁₀(P value) > 2.0 reveals the overlap in protein alterations observed within ΔpglL, denoted in blue, within mutants lacking MotB, FliF, and BCAM0505. (B) Heatmap of fold changes in proteins observed altered within ΔpglL versus WT reveals similarities in the directionality of protein alterations observed within ΔfliF and ΔmotB strains. (C) Enrichment analysis demonstrates a high degree of overlap within the proteins altered within ΔpglL versus WT compared to ΔmotB versus WT and ΔfliF versus WT. (D) Enrichment analysis of gene ontology (biological process) terms enriched within protein alterations observed within ΔfliF.

findings support that the loss of *fliF* and *motB* results in similar proteomic impacts as those observed within ΔpglL. To understand the importance of glycosylation for the function of *fliF* and *motB* these proteins were selected for further analysis using site-directed mutagenesis.

## Glycosylation at S₃₅₈ within FliF and S₃₃₁ within MotB are dispensable for motility

The observed proteomic changes within ΔmotB and ΔfliF, as well as the loss of these proteins in the absence of PglL suggests glycosylation may influence the stability and/or functionality of these proteins. In light of our previous work demonstrating reductions in motility associated with loss of glycosylation (41, 55) as well as FliF/MotB homologs being known to be associated with flagella-based motility (99), we confirmed the requirement of FliF and MotB for motility within *B. cenocepacia* (Fig. 5A and B). Plasmid-based complementation using the inducible vector pSCrhaB2 (71) allowed the partial restoration of motility within ΔfliF pSCrhaB2-*fliF* upon induction, while motility was restored within ΔmotB pSCrhaB2-*motB* without induction (Fig. 5A and B). Surprisingly,

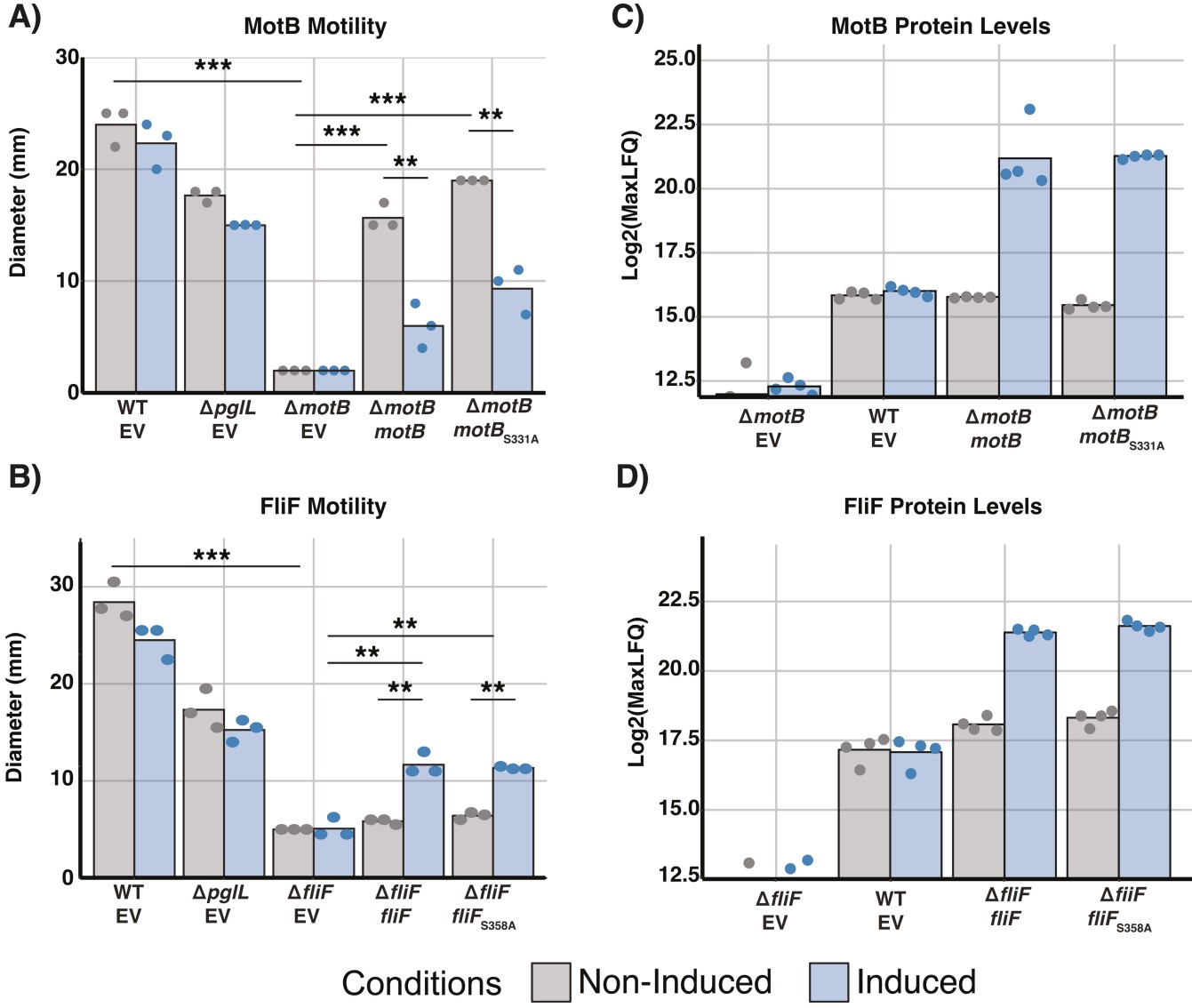

**FIG 5** Complementation of Δ*fliF* and Δ*motB* partially restores motility and protein levels. (A and B) Motility assays of complemented Δ*fliF* and Δ*motB* demonstrate the restoration of motility by the expression of wild-type *fliF* or *motB* regardless of the absence or presence of the known glycosylation sites $S_{358}$ and $S_{331}$. (C and D) Proteomic analysis confirms the restoration of MotB and FliF protein levels within complemented strains. Induced samples are denoted by blue boxes, while non-induced samples are denoted by gray boxes. The plasmid contained within each strain is shown below the strain name. EV—empty pSCrhaB2, *motB*—pSCrhaB2-*motB*, *motB*$_{S331A}$—pSCrhaB2-*motB*$_{S331A}$, *fliF*—pSCrhaB2-*fliF*, and *fliF*$_{S358A}$—pSCrhaB2-*fliF*$_{S358A}$ with the lower boundaries of the MaxLFQ shown defined by the imputed values observed within Δ*fliF* and Δ*motB* EV, respectively.

within Δ*motB* pSCrhaB2-*motB* induction reduced motility (Fig. 5B) with reductions in motility also observed within the parental strain *B. cenocepacia* K56-2 in response to MotB induction (Fig. S20), suggesting the overexpression of MotB is detrimental to motility. To assess *motB/fliF* complementation, proteomic analysis was undertaken (Tables S8 and S9; Fig. S21 and 22) confirming the restoration of MotB with pSCrhaB2-*motB* in Δ*motB* and FliF with pSCrhaB2-*fliF* in Δ*fliF* (Fig. 5C and D). Interestingly despite the observation of near wild-type levels of FliF even without induction by proteomic analysis, induction was required to partially restore motility within Δ*fliF* (Fig. 5B and D). The proteomic analysis supports that while FliF is restored without induction additional motility-associated proteins including MotA (BCAL0126) and MotB within Δ*fliF* were only restored, albeit partially, with induction (Fig. S21). To assess the role of glycosylation sites within FliF and MotB, Alanine substitutions were created based on known glycosylation

sites (FliF$_{S358A}$ and MotB$_{S331A}$, Table S2), and the impact on motility was examined. As above, complementation of Δ*motB* with pSCrhaB2-*motB$_{S331A}$* restored motility even without induction (Fig. 5A and B), while Δ*fliF* with pSCrhaB2-*fliF$_{S358A}$* only restored motility with induction (Fig. 5C and D). Combined these results support while FliF and MotB are required for motility the known *O*-linked glycosylation sites of these proteins appear dispensable for motility.

## FliF and MotB contribute to motility independent of their glycosylation status

Finally, to confirm the loss of glycosylation within FliF$_{S358A}$ and MotB$_{S331A}$ as well as investigate the requirement of these proteins for motility in the absence of glycosylation, we created Δ*pglL*Δ*fliF* and Δ*pglL*Δ*motB*. Consistent with the requirement of FliF and MotB for motility within glycosylation null backgrounds we observe marked reductions in motility within both Δ*pglL*Δ*fliF* and Δ*pglL*Δ*motB* compared to wild-type and Δ*pglL* strains with a partial restoration in motility observed upon complementation with pSCrhaB2-*fliF*/pSCrhaB2-*motB* respectively (Fig. 6A and B). Consistent with complementation Western analysis reveals comparable levels of FliF/MotB upon induction within Δ*fliF*/Δ*motB* compared to Δ*pglL*Δ*fliF*/Δ*pglL*Δ*motB,* respectively (Fig. 6C and D). Analysis of FliF$_{S358A}$ and MotB$_{S331A}$ expressed within Δ*fliF*/Δ*motB* strains demonstrates that FliF$_{S358A}$/MotB$_{S331A}$ possess comparable sizes to FliF/MotB expressed within

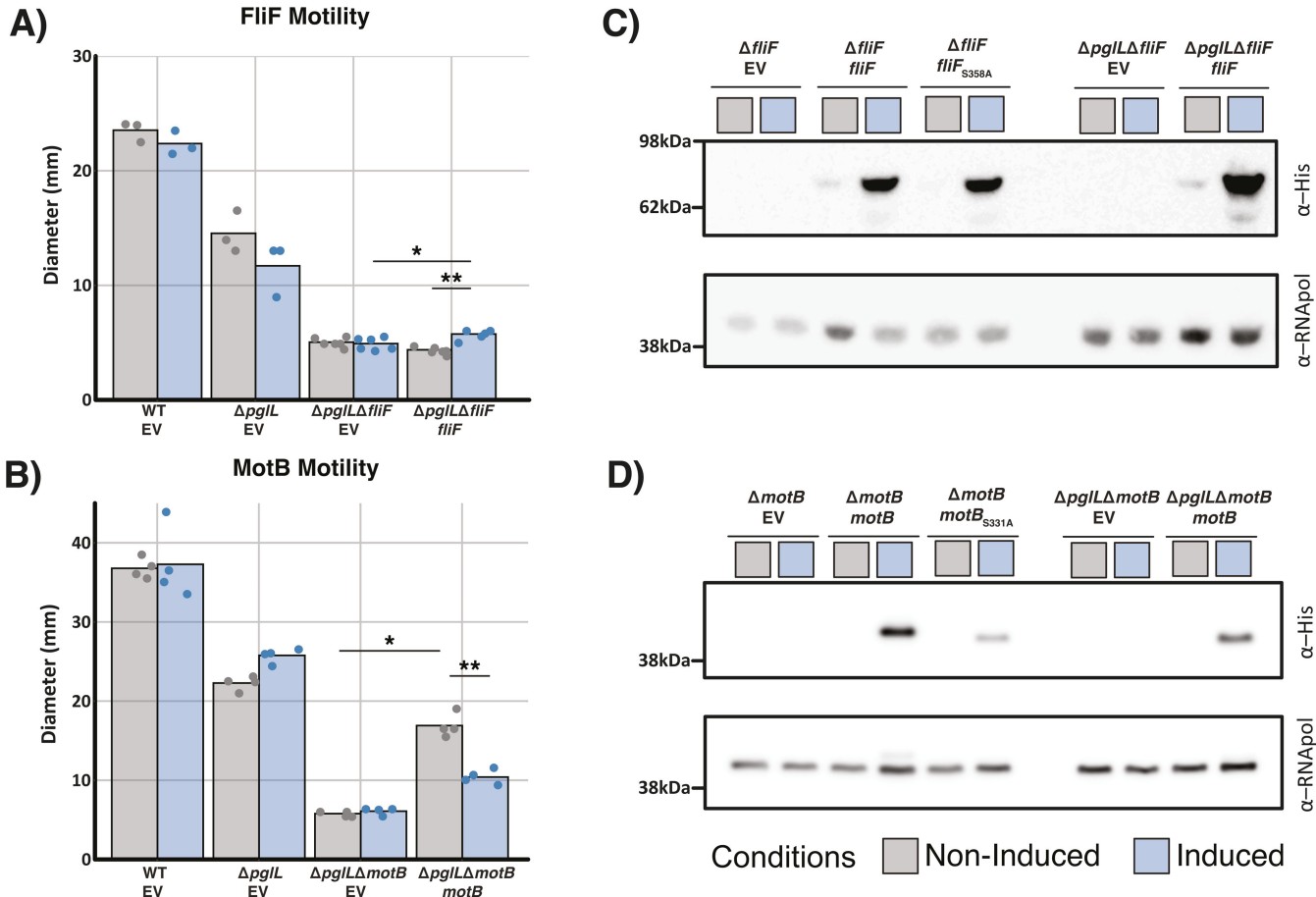

**FIG 6** FliF and MotB contribute to motility in the absence of glycosylation. (A and B) Motility assays of complemented Δ*pglL*Δ*fliF* and Δ*pglL*Δ*motB* compared to WT and Δ*pglL* demonstrate the restoration of motility by the expression of wild-type *fliF* or *motB* compared to EV. (C and D) Western analysis complemented strains demonstrate mobility shifts with FliF or MotB Alanine substitutions as well as FliF or MotB expressed within Δ*pglL* backgrounds. Induced samples are denoted by blue boxes, while non-induced samples are denoted by gray boxes. The plasmid contained within each strain is shown below the strain name. EV—empty pSCrhaB2, *motB*—pSCrhaB2-*motB*, *motB$_{S331A}$*—pSCrhaB2-*motB$_{S331A}$*, *fliF*—pSCrhaB2-*fliF*, and *fliF$_{S358A}$*—pSCrhaB2-*fliF$_{S358A}$*.

glycosylation null backgrounds supporting the abolishment of glycosylation within these point-mutants (Fig. 6C and D). Additionally, we also note that the complementation of Δ*pglL* with pSCrhaB2-*motB*/pSCrhaB2-*fliF* is insufficient to restore motility defects associated with Δ*pglL* (Fig. S23). Combined these results support the loss of glycosylation within FliF$_{S358A}$ and MotB$_{S331A}$ and highlight that these proteins contribute to motility independent of their glycosylation status.

## DISCUSSION

Within this study, we have expanded and curated the known glycoproteome of *B. cenocepacia* revealing at least 141 proteins or ~2% of the proteome is subjected to glycosylation [Supplementary Data 1 and 2 (https://doi.org/10.6084/m9.figshare.25492774.v1)] and, by leveraging this resource assessed the impact of glycosylation on two proteins associated with motility in *B. cenocepacia,* FliF and MotB. While several studies (55, 100, 101) have explored the impact of glycosylation on the broader proteomes, the dynamics of bacterial glycoproteomes, both across growth phases and in response to the loss of glycosylation, have largely been unexplored due to the technical challenges associated with tracking bacterial glycoproteins. The recent refinements to MS approaches such as DIA analysis (57, 58), now enable quantitative assessments of bacterial proteomes to be assessed at scale. While MS approaches to undertake such studies are now widely accessible, to our knowledge, only a single study within *Campylobacter jejuni* has leveraged similar approaches to specifically track glycoprotein occupancy in response to the loss of *N*-linked glycosylation utilizing these insights to uncover previously unrecognized glycosylation sites (102). The recent identification of hundreds of glycosylation events within several Gram-negative species (102–105) highlights the need for robust tools to track bacterial glycoproteomes with this work highlighting how dynamic information can be used to prioritize the identification of potentially important glycoproteins and sites.

The identification of only a modest increase in the known glycoproteome (<10%, Fig. 1) across differing culturing conditions as well as growth phases coupled with our measurements of >100 glycoproteins across these conditions (Fig. 2) supports the notion that the glycoproteome is a constitutive and invariant feature of the *B. cenocepacia* proteome. Consistent with this, the enzymes associated with *O*-linked glycosylation are produced at similar protein levels across all growth phases/culturing media examined within this study (Fig. S8). Across the 141 glycoproteins confirmed within *B. cenocepacia*, we found that despite large proteomic changes, the absence of glycosylation only impacts the abundance of five observable glycoproteins (Fig. 3; Fig. S11). While this work recapitulates our previous observations of the loss of BCAL1086 and BCAL2974 (55), it supports the loss of glycoproteins in response to the disruption of glycosylation are the exception, not the rule within the *B. cenocepacia* glycoproteome. Consistent with this concept, recent work on *C. jejuni N*-linked glycosylation has demonstrated that glycosylation events can be required for the stabilization of protein complexes exerting their effects independent of changes in protein levels (106). These observations support the assessments of protein properties invisible to standard proteomic assays, such as exploring changes within the interactome, or protein folding, are likely required to dissect the underlying alterations driving the large proteomic impact observed by the loss of glycosylation.

Within this work, we explored the link between *B. cenocepacia* glycosylation, and the reduced motility observed within Δ*pglL* (41, 55). Combining our glycoproteome and proteomic data sets, we identified two motility-associated glycoproteins, MotB and FliF, that are dramatically reduced in response to the loss of glycosylation with the disruption of these proteins observed to drive proteomic effects mirroring those seen within Δ*pglL* (Fig. 3 and 4). However, our analysis of the role of glycosylation within these proteins using site-directed mutagenesis and glycosylation null strains demonstrates that glycosylation does not appear to directly impact the role of these proteins (Fig. 5 and 6). Interestingly, while we did observe complementation within the Δ*pglL*Δ*fliF*

and Δ*pglL*ΔmotB backgrounds, the observed levels of motility restoration were reduced compared to complementation within the Δ*fliF* and Δ*motB* backgrounds (Fig. 5 and 6). We hypothesize that the widespread proteomic impacts observed from the loss of O-linked glycosylation may reduce the amount of functional FliF and MotB despite complementation, leading to similar protein levels between Δ*pglL*Δ*fliF* and Δ*fliF*, as well as between Δ*pglL*Δ*motB* and Δ*motB* (Fig. 6). Regardless, our observations suggest that the *O*-linked glycosylation sites of S358 and S331 within FliF and MotB, respectively, are not integral for the function of these glycoproteins in motility. While this does not exclude the potential importance of these sites for functions beyond motility, it is notable that our proteomic analysis supports no clear proteomic differences from the absence or presence of these glycosylation events (Tables S8 and S9; Fig. S21 and S22). However, it should be noted that the observation of the dispensable nature of these sites may only occur under the ideal conditions examined here which are dramatically different from *B. cenocepacia's* natural reservoir. This said, within *C. jejuni*, it has been noted that multiple *N*-linked glycoproteins possess glycosylation sites that are dispensable for protein function such as Cj1496c (107) and Cj0143c (108). In *C. jejuni*, site-directed mutagenesis of glycosylation events in Cj1496c and Cj0143c demonstrate glycosylation sites in these proteins do not impact chicken colonization, yet loss of these proteins is detrimental to colonization (107, 108). These findings highlight that the presence of glycosylation in bacterial systems alone is not an indicator of the importance of a given glycosylation event and that the glycosylation events of FliF and MotB appear dispensable within *B. cenocepacia*.

In summary, this work furthers our understanding of the breath and dynamics of *Burkholderia* glycoproteins revealing the *B. cenocepacia* glycoproteome composition is constitutively expressed and largely invariant within the proteome. Using DIA proteomics, we demonstrate that while the loss of glycosylation drives large proteome alterations in *B. cenocepacia* few glycoproteins display significant alterations in abundance. These observations support the loss of glycosylation likely impacts protein properties invisible to direct proteomic analysis, such as protein interactions. Combined our identification of 141 glycoproteins within *B. cenocepacia* represents the largest known PglL-mediated O-linked glycoproteome identified in Gram-negative species to date providing a roadmap for systematic characterization of *B. cenocepacia* glycoproteins to understand the role of these proteins in physiology and virulence.

## ACKNOWLEDGMENTS

The authors thank the Melbourne Mass Spectrometry and Proteomics Facility of The Bio21 Molecular Science and Biotechnology Institute for access to MS instrumentation.

N.E.S. is supported by an Australian Research Council Future Fellowship (FT200100270) and was supported by an ARC Discovery Project Grant (DP210100362).

## AUTHOR AFFILIATION

[1]Department of Microbiology and Immunology, University of Melbourne at the Peter Doherty Institute for Infection and Immunity, Melbourne, Australia

## PRESENT ADDRESS

Jessica M. Lewis, School of Life Sciences, University of Warwick, Coventry, United Kingdom
Pauline M. L. Coulon, Australian Institute for Microbiology and Immunology, Faculty of Science, University of Technology Sydney, Ultimo, Australia

## AUTHOR ORCIDs

Nichollas E. Scott (iD) http://orcid.org/0000-0003-2556-8316

## FUNDING

| Funder | Grant(s) | Author(s) |
|---|---|---|
| Australian Research Council | FT200100270, DP210100362 | Nichollas E. Scott |

## DATA AVAILABILITY

Mass spectrometry data (RAW files, FragPipe outputs, Spectronaut outputs and experiment files, MaxQuant search outputs, Rmarkdown scripts, and output tables) have been deposited into the PRIDE ProteomeXchange repository (109, 110). All PRIDE accession numbers, descriptions of the associated experiments and experiment types (DIA or DDA) are provided within Table S13. Supplementary Data set 1 and 2 have been deposited within Figshare (https://doi.org/10.6084/m9.figshare.25492774.v1). All pride proteomic datasets are available at accessions PXD043184, PXD043186, PXD043187, PXD043188, PXD043193, PXD043194, PXD043197, PXD043221, PXD043227, PXD043280, and PXD045246.

## ADDITIONAL FILES

The following material is available online.

### Supplemental Material

**Supplemental material (Spectrum00346-24-S0001.pdf).** Tables S10 to S13; Fig. S1 to S24.
**Table S1 (Spectrum00346-24-S0002.xlsx).** Glycopeptide identifications within *B. cenocepacia* K56-2 identified across growth conditions using FAIMS and ZIC-HILIC enrichments.
**Table S2 (Spectrum00346-24-S0003.xlsx).** Glycoproteins identified across *B. cenocepacia*.
**Table S3 (Spectrum00346-24-S0004.xlsx).** DIA Protein level analysis of *B. cenocepacia* K56-2 grown in LB, TSB and ASM culturing media (Stationary and Log growth phases).
**Table S4 (Spectrum00346-24-S0005.xlsx).** DIA Protein level analysis of *B. cenocepacia* K56-2 wildtype, ΔpglL and ΔpglLΔamrAB::native-promotor-pglL-his grown to stationary phase in LB, TSB and ASM culturing media.
**Table S5 (Spectrum00346-24-S0006.xlsx).** DDA protein level LFQ analysis of *B. cenocepacia* ΔmotB candidates; ΔfliF candidates and ΔBCAM0505 candidates compared to WT at stationary phase in LB.
**Table S6 (Spectrum00346-24-S0007.xlsx).** DIA Protein level analysis of ΔmotB, ΔfliF, ΔBCAM0505 and ΔpglL compared to WT at stationary phase in LB.
**Table S7 (Spectrum00346-24-S0008.xlsx).** Enrichment analysis of proteome changes observed within ΔmotB, ΔfliF and ΔpglL compared to WT at stationary phase in LB.
**Table S8 (Spectrum00346-24-S0009.xlsx).** DIA Protein level analysis of ΔmotB and WT containing pScrhaB2, pSCrhaB2-motB or pSCrhaB2-motBS331A with / without induction (0.1% rhamnose) at stationary phase in LB.
**Table S9 (Spectrum00346-24-S0010.xlsx).** DIA Protein level analysis of ΔfliF containing pScrhaB2, pSCrhaB2-fliF or pSCrhaB2-fliFS358A and WT containing pScrhaB2 with / without induction (0.1% rhamnose) at stationary phase in LB.

### Open Peer Review

**PEER REVIEW HISTORY (review-history.pdf).** An accounting of the reviewer comments and feedback.

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
