## [Reviewer comments · Microbiology Spectrum]

Microbiology Spectrum

Glycoproteomic and Proteomic analysis of *Burkholderia cenocepacia* reveals glycosylation events within *FliF* and *MotB* are dispensable for motility.

Jessica Lewis, Leila Jebeli, Pauline Coulon, Catrina Lay, and Nichollas Scott

Corresponding Author(s): Nichollas Scott, The University of Melbourne

Review Timeline:

Submission Date:	February 10, 2024
Editorial Decision:	March 6, 2024
Revision Received:	March 29, 2024
Editorial Decision:	April 8, 2024
Revision Received:	April 9, 2024
Accepted:	April 16, 2024

Editor: Silvia Cardona

Reviewer(s): Disclosure of reviewer identity is with reference to reviewer comments included in decision letter(s). The following individuals involved in review of your submission have agreed to reveal their identity: Jennifer Geddes-McAlister (Reviewer #1); Bart Devreese (Reviewer #2)

Transaction Report:

DOI: <https://doi.org/10.1128/spectrum.00346-24>

Re: Spectrum00346-24 (Glycoproteomic analysis of Burkholderia cenocepacia reveals the flagella apparatus proteins FliF and MotB are glycosylated yet glycosylation is dispensable for motility.)

Dear Prof. Nichollas E Scott:

Thank you for submitting your manuscript to Microbiology Spectrum. Two experts in the field have reviewed your article. The reviewers found your manuscript very well written, with robust data presentation and analysis. Additionally, they have provided comments to improve the quality of your work.

Their recommendations are provided below.

Revision Guidelines

Sincerely,
Silvia Cardona
Editor
Microbiology Spectrum

Reviewer #1 (Comments for the Author):

General comments

The manuscript is well written and well presented. The relevance and approach used are robust and the methods are provided with sufficient detail and statistical analysis of proteomics and glycoproteomics datasets is robust. An excellent manuscript with conclusions well supported by the results and figures.

Specific comments

Line 49 - define CF

Line 124 - the sentence is a little awkward. Consider revising.

Line 165 - write out E.coli in full for first introduction

Line 514 (and throughout) - why are the three strains selected and generated (i.e., Δ motB, Δ fliF, and Δ BCAM0505) instead of all five of the glycoproteins. Justification is needed from the beginning.

Line 515 - given the extensive information provided with the supplemental tables for the proteome assessment of the mutant vs. wild-type strains and the rather non-traditional approach to confirming a mutant (i.e., via proteomics) and not genetic approaches, I suggest including the fold-change or LFQ intensity values (to demonstrate absence or below the limit of detection for the protein within the mutant strain) within the main text. The values in the supplemental table appear to be imputed and so confirmation that the gene (and protein) are absent by LFQ is needed. Notably, the imputation is well represented within Supp. Fig. 13.

Line 555 - why confirm mutants and restoration by proteomics and not WGS to ensure that additional genetic disruptions were not made that may influence the changes in proteome remodeling upon comparison to wild-type?

Fig. 2 D-E - the labels on the red dots are too small to read - value in including them?

Reviewer #2 (Comments for the Author):

Glycosylation is quite recently recognized as a common post-translational modification in bacteria. Most studies were initiated in *Campylobacter jejuni*, but today it is understood that this modification is widespread. Burkholderia belong to the species that are most studied in this context, but the exact size of the glycoproteome and its function is not well understood.

In this work, Lewis et al. used state-of-the-art mass spectrometric technology to perform a comprehensive analysis of the glycoproteome of species from the Burkholderia cenocepacia complex, known as an opportunistic pathogens to infect people with cystic fibrosis and related immunocompromised individuals. Not only do they provide data about the in depth identification of glycoproteins, they also used knock outs in the glycosylation machinery to investigate the impact of glycosylation on those proteins' abundance. Strikingly, while the global proteome undergoes a somewhat unexpected large change in abundance, proteins that are glycosylated do not change in abundance, hinting that glycosylation does not have an impact on their stability. Exceptions herein are the previously known glycoproteins MotB and FliF involved in motility. Additional studies on knock outs for the corresponding genes or specific mutation of their glycosite indicate are studies, revealing that in contrast what was believed before, that glycosylation of MotB and FliF has no impact on motility.

While the latter may seem a negative result, together with the unprecedented powerful and comprehensive analysis of the overall Burkholderia cenocepacia glycoproteome and report of the impact of loss of glycosylation in general, it sheds new lights on the function of bacterial glycosylation.

While the overall setup, methodology and the quality of data and reporting of results are excellent, there are some items that should be better clarified or improved, as well as some additional information might be highly useful. Below these are listed in chronological order of appearance in the text.

1. At some instances, the mass spectrometric data are searched against the J2315 database, whereas in other analyses both K56-2 and J2315 proteome databases are searched. Since the former contains more proteins, why has the glycopeptide identification not been performed in this database. This reviewer is aware that Uniprot merged most K56-2 entries in the J2315 database, but still it is known that the K56-2 strain is genetically somewhat different. It is also not clear whether the two databases were merged or whether they were individually searched for the DIA and DDA quantitative analyses. This should be clarified in the method section.

2. It should be noted when the Uniprot databases were uploaded. Recently, the annotations of the Burkholderia J2315 proteins improved, with more previously hypothetical/putative proteins being annotated. This particularly refers to the statements at page 17 saying that half of the discovered glycoproteins were unknown. Did the authors make an effort to manually annotated these proteins. More specifically, concerning the lack of GO ontology terms, it could have been of interest to investigate the presence of signal peptides or membrane anchors that could point to cellular location as the glycosylation machinery is supposed to be periplasmic. Are all identified proteins supposed to be periplasmic/ membrane oriented or secreted ?

3. The method section should be checked for completeness, in particular for the database searching. For example, supplemental data refer to Byonic searches, but this is not included in the methodology.

4. In this respect, the authors are invited to provide some extra information on the different ms methods and data analysis approaches that are used. This manuscript is submitted to a more general microbial journal and the approaches and result reports require high level knowledge of proteomics methodologies generally not known to microbiologists.

5. Figure 1A is somewhat obsolete as the same information can be easily deduced from Figure 1B

6. A minor remark : the glycan marker ions in figure 1E are in the same color as c ions. This should be avoided.

7. Figure 4A is in its present form not easy to read. In particular the grayscale used in the bottom part where correlation lines are drawn is not visible in a print on this scale.

8. With respect to Figure 4D, the enrichment analysis is done on the deltaFliF strain. While there are certainly similarities

between this strain and the Δ pglL strain, as shown from the heat map in figure 4B, there are some pertinent differences. It deserves to be demonstrated that the same enrichment for flagellum and chemotaxis proteins to be affected by FliF mutations are also true in the glycosylation mutant. At first sight, there seem to be some discrepancies in fold changes specifically of proteins involved in flagellin formation that can be easily linked to the *fliF* gene deletion. But the impact of glycosylation machinery deletion outside the glycoproteome is not as such evaluated at the GO level. This could hint to some non-obvious impact of loss of glycosylation as discussed on page 24

Response to Reviewers

Response to reviewers are denoted in red

Reviewer comments

Reviewer: 1

General comments

The manuscript is well written and well presented. The relevance and approach used are robust and the methods are provided with sufficient detail and statistical analysis of proteomics and glycoproteomics datasets is robust. An excellent manuscript with conclusions well supported by the results and figures.

We thank the reviewer for their positive and supportive comments on the manuscript.

Specific comments

Line 49 - define CF

We thank the reviewer for highlighting this oversight and have replaced CF with Cystic Fibrosis within the Importance statement.

Line 124 - the sentence is a little awkward. Consider revising.

We thank the reviewer for their suggestion, we have altered the sentence (Page 6, Line 137):

“Combined these advancements now allow DIA to provide deep proteomic coverage and the ability to precisely track protein classes traditionally challenging to monitor including microproteins (<100 amino acid in length)⁵⁷ as well as low abundant proteins^{52,53}.”

to

“Combined these advancements now allow DIA to provide deep proteomic coverage and track even challenging protein classes such as microproteins (<100 amino acid in length)⁵⁷ as well as low abundant proteins^{52,53}.”

Line 165 - write out E.coli in full for first introduction

We thank the reviewer for flagging this oversight, the full name, *Escherichia coli* has been added to line 165 (now line 177)

Line 514 (and throughout) - why are the three strains selected and generated (i.e., Δ motB, Δ fliF, and Δ BCAM0505) instead of all five of the glycoproteins. Justification is needed from the beginning.

We thank the reviewer for pointing out this oversight regarding the prioritization of glycoproteins for characterization. Out of the five glycoproteins [BCAL1086, BCAL2974, FliF

(BCAL0525), BCAM0505, and MotB (BCAL0127)] that exhibited changes in response to the loss of glycosylation, we have previously investigated the proteomic effects of the disruption of *BCAL1086* and *BCAL2974* in our prior (and more limited) proteomic study on how the absence of glycosylation affects *B. cenocepacia* K56-2 (Oppy *et al.* mSphere 2019). As such, within this study we focused on the glycoproteins newly identified to be sensitive to the loss of glycosylation from our DIA analysis [FliF (BCAL0525), BCAM0505, and MotB (BCAL0127)]. To clarify this point we have now added the following sentence to the results section (Page 21, Line 532):

“As our previous studies have demonstrated that the disruption of *BCAL1086* or *BCAL2974* does not drive the proteome changes observed in $\Delta pgII$ ⁵⁵ we focused on the glycoproteins MotB, FliF, and BCAM0505.”

Line 515 - given the extensive information provided with the supplemental tables for the proteome assessment of the mutant vs. wild-type strains and the rather non-traditional approach to confirming a mutant (i.e., via proteomics) and not genetic approaches, I suggest including the fold-change or LFQ intensity values (to demonstrate absence or below the limit of detection for the protein within the mutant strain) within the main text. The values in the supplemental table appear to be imputed and so confirmation that the gene (and protein) are absent by LFQ is needed. Notably, the imputation is well represented within Supp. Fig. 13.

Our proteomic analysis of strains $\Delta motB$, $\Delta fliF$, and $\Delta BCAM0505$ support that these three proteins are undetectable within mutants compared to wild type which is the key point highlighted in Supplementary Figures 13B, 14B and 15B yet we appreciate that this point has not clearly been made. To improve data transparency, we have now merged Supplementary tables 6, 7 and 8, for now known as Supplementary table 5, including the non-imputed data as well as updated the text of the main document to state (Page 20, Line 534):

“To assess if the loss of *motB*, *fliF*, and BCAM0505 contributes to the proteomic impact seen in the absence of glycosylation, we generated $\Delta motB$, $\Delta fliF$, and $\Delta BCAM0505$ strains, and using DDA proteomic analysis confirmed the loss of these three proteins within mutants (Supplementary figures 13 to 15, Supplementary tables 5).”

It also should be noted that genetic approaches to confirm all mutants were undertaken, to make this point clearer we have now updated Page 10, Line 214 to 216 to state:

“Mutagenesis was confirmed using screening oligonucleotides which bind outside the region of recombination (Supplementary Document Table 3) using GoTaq DNA polymerase-based PCR screening supplemented with 10% DMSO.”

Line 555 - why confirm mutants and restoration by proteomics and not WGS to ensure that additional genetic disruptions were not made that may influence the changes in proteome remodeling upon comparison to wild-type?

We appreciate our use of proteomics to confirm mutants is unorthodox and admit our use of this approach is driven by our access to this technology. Within the lab after confirmation of mutagenesis using PCR based approaches our first pass to validate mutants, especially strains for use in large studies (such as multi-strain comparisons) is to undertake proteomics of independent mutants compared to WT strains. This approach does not replace our molecular (PCR based) approaches of confirming mutations but is intended to provide further and orthogonal confirmation of deletions. It should be noted with the increasing accessibility of WGS we are beginning to integrate this into our strain validation workflows, yet these strains have not been subjected to WGS. Additionally, complementation was observed to restore phenotypes, further supporting that the phenotypes assigned to our mutants are the result of the disruption in question.

Fig. 2 D-E - the labels on the red dots are too small to read - value in including them?

The reviewer raises a fair point, while we initially provided the individual measurements as red dots for data transparency, we agree that due to the complexity of this figure this does make these data points extremely small. As such we have updated the figure to use error bars in place of individual data points:

Updated Figure 2.

Reviewer #2 (Comments for the Author):

Glycosylation is quite recently recognized as a common post-translational modification in bacteria. Most studies were initiated in *Campylobacter jejuni*, but today it is understood

that this modification is widespread. Burkholderia belong to the species that are most studied in this context, but the exact size of the glycoproteome and its function is not well understood.

In this work, Lewis et al. used state-of-the-art mass spectrometric technology to perform a comprehensive analysis of the glycoproteome of species from the Burkholderia cenocepacia complex, known as an opportunistic pathogens to infect people with cystic fibrosis and related immunocompromised individuals. Not only do they provide data about the in depth identification of glycoproteins, they also used knock outs in the glycosylation machinery to investigate the impact of glycosylation on those proteins' abundance. Strikingly, while the global proteome undergoes a somewhat unexpected large change in abundance, proteins that are glycosylated do not change in abundance, hinting that glycosylation does not have an impact on their stability. Exceptions herein are the previously known glycoproteins MotB and FliF involved in motility. Additional studies on knock outs for the corresponding genes or specific mutation of their glycosite indicate are studies, revealing that in contrast what was believed before, that glycosylation of MotB and FliF has no impact on motility. While the latter may seem a negative result, together with the unprecedented powerful and comprehensive analysis of the overall Burkholderia cenocepacia glycoproteome and report of the impact of loss of glycosylation in general, it sheds new lights on the function of bacterial glycosylation.

While the overall setup, methodology and the quality of data and reporting of results are excellent, there are some items that should be better clarified or improved, as well as some additional information might be highly useful. Below these are listed in chronological order of appearance in the text.

We thank the reviewer for their kind and supportive comments on the manuscript.

1. At some instances, the mass spectrometric data are searched against the J2315 database, whereas in other analyses both K56-2 and J2315 proteome databases are searched. Since the former contains more proteins, why has the glycopeptide identification not been performed in this database. This reviewer is aware that Uniprot merged most K56-2 entries in the J2315 database, but still it is known that the K56-2 strain is genetically somewhat different. It is also not clear whether the two databases were merged or whether they were individually searched for the DIA an DDA quantitative analyses. This should be clarified in the method section.

The review raises an excellent point on the databases used for analysis.

For glycopeptide focused analysis the J2315 database was used in line with our previous glycoproteomic studies (Ahmad Izaham *et al.* Mol Cell Proteomics. 2020; Ahmad Izaham *et al.* J Proteome Res. 2021) yet we appreciate that this may lead to the loss of some glycopeptides due to variations in the peptide sequences between J2315 and K56-2. While our previous analysis of site conservation suggests that at the peptide level there is very little alteration between strains within the region flanking glycosylation sites (>95% identity across 294 *B. cenocepacia* genomes within 69 sites, see Hayes *et al.* Commun Biol. 2021; 4: 1045.) there may be cases where glycopeptides are missed due to sequence variation. This

does not undermine the novel glycopeptides identified but may mean additional glycoproteins could be identified if the K56-2 database was used to assess these glycoproteomic datasets. To ensure readers are aware of this caveat we have now added the following sentence (Page 18, Line 453):

“Glycopeptides were identified against the J2315 proteome with our previous work highlighting the high sequence identity within the surrounding amino acids of glycosylation sites within *B. cenocepacia*³⁸, however this approach is insensitive to the detection of glycopeptides unique to K56-2.”

For DIA and DDA quantitative analyses both the J2315 and K56-2 databases were searched together, effectively merging these databases. By searching these databases together information on the identified proteins/peptides are assigned to both J2315 and K56-2 assignments which streamlines the matching of proteins to known J2315 accessions. This point has now been highlighted in (Page 15, line 363):

“Data files were searched against the *B. cenocepacia* K56-2 proteome (Uniprot accession: UP000011196, 7467 proteins, downloaded October 13th 2020)⁹⁴ and *B. cenocepacia* strain J2315 (Uniprot accession: UP000001035)⁹⁰, effectively merging these proteomes enabling the matching of proteins to both J2315 and K56-2 accessions”

2. It should be noted when the Uniprot databases were uploaded. Recently, the annotations of the Burkholderia J2315 proteins improved, with more previously hypothetical/putative proteins being annotated. This particularly refers to the statements at page 17 saying that half of the discovered glycoproteins were unknown. Did the authors make an effort to manually annotated these proteins. More specifically, concerning the lack of GO ontology terms, it could have been of interest to investigate the presence of signal peptides or membrane anchors that could point to cellular location as the glycosylation machinery is supposed to be periplasmic. Are all identified proteins supposed to be periplasmic/membrane oriented or secreted ?

We thank the reviewer for the suggestion to clearly outline when the Uniprot databases used in this study were downloaded, we have now provided these dates adding these details to Page 14, line 346:

“*B. cenocepacia* reference proteome J2315 Uniprot accession: UP000001035, 6,993 proteins, downloaded July 24th 2020”

And to Page 15, line 365:

“*B. cenocepacia* K56-2 proteome (Uniprot accession: UP000011196, 7467 proteins, downloaded October 13th 2020”

Regarding our statement on Page 17 we apologize to the reviewer for our poor choice of wording. The glycoproteins highlighted here as unknown do not correspond to open reading frames which were previously unassigned within the J2315 proteome but to known proteins

that were unrecognized as glycosylated. To highlight this point, we have modified the sentence on (Page 18, Line 467) from:

“revealing 12 of the 54 glycoproteins identified within this study correspond to previously unknown *B. cenocepacia* glycoproteins (Figure 1D, Supplementary Figure 3).”

to

“revealing 12 of the 54 glycoproteins identified within this study correspond to previously unrecognized *B. cenocepacia* glycoproteins (Figure 1C, Supplementary Figure 3).”

Based on the periplasmic location of the oligosaccharyltransferase all protein subjected to glycosylation are exposed to the periplasmic space. To help summarise the details on the predicted number of transmembrane domains, the presence of signal peptides as well as the GO terms assigned to proteins of the known glycoproteome, we have now updated Supplementary table 2 to contain these details. Consistent with the expectation that proteins are secreted we observe most glycoproteins contain transmembrane domains, a Signal tag or are assigned as membrane associated proteins based on GO terms. We have now highlighted that these details are provided in Supplementary Table 2, page 27 line 704:

“Supplementary Table 2. Glycoproteins identified across *B. cenocepacia*. Curated list of all known glycoproteins within *B. cenocepacia* including confirmed glycosylation sites, residue of the assigned glycosylation site and if the glycoproteins have been previously identified within previous studies. For each glycoprotein the location and number of predicted transmembrane domains, location of the Signal tag and GO terms associated with each protein are provided derived from the *B. cenocepacia* J2315 proteome (UP000001035).”

3. The method section should be checked for completeness, in particularly for the database searching. For example, supplemental data refer to Byonic searches, but this is not included in the methodology.

We apologise to the reviewer for the confusion on the origins of the Byonic glycopeptide assignments. To ensure the glycopeptides/glycoproteins identified within this study were truly unique we curated a list of all known glycoproteins within *B. cenocepacia* identified across previous studies (Supplementary table 2). During this curation to ensure only true glycoproteins were included in this final list we manually validated all unique assignments which had not been previously manually validated in the studies Ahmad Izaham *et al.* Mol Cell Proteomics. 2020 and Ahmad Izaham *et al.* J Proteome res. 2021 where glycopeptides were identified using Byonic. To clearly highlight the origin of these glycopeptide assignments we have added the following sentences to the method section (page 14, line 354):

“To assess the uniqueness of identified glycoproteins within this study we curated a list of all previously known *B. cenocepacia* glycoproteins^{38,39,41,43}. Unique glycopeptides identified using the glycopeptide identification tool Byonic (Protein Metrics Inc.⁹³) yet not previously manually confirmed from studies^{39,43} were annotated to ensure correctness resulting in a

total of 129 unique glycoproteins across all studies^{38,39,41,43} (Supplementary Table 2) with unique glycopeptides from these previous Byonic searches provided within Supplementary Data 2.”

4. In this respect, the authors are invited to provide some extra information on the different ms methods and data analysis approaches that are used. This manuscript is submitted to a more general microbial journal and the approaches and result reports require high level knowledge of proteomics methodologies generally not known to microbiologists.

We thank the reviewer for this suggestion, and we have now expanded the introduction to provide more information and background to readers on the MS methods used particularly glycoproteomic analysis, Page 5, Line 114:

“The analysis of bacterial proteomes using liquid chromatography–mass spectrometry (LC-MS) has emerged as the quintessential approach for the quantitation of microbial proteomes and post-translational modifications^{44,45}. Within bottom-up proteomics, the most widely implemented form of proteomics, proteins are first digested into peptides prior to analysis⁴⁶. Once digested, peptides can be analysed directly or subjected to enrichment approaches to enhance the detection of specific peptides of interest, such as glycopeptides in the case of glycoproteomics⁴⁷. In glycoproteomics a range of enrichment approaches can be used to isolate glycopeptides⁴⁷ with Zwitterionic hydrophilic interaction liquid chromatography (ZIC-HILIC)^{38,39,41,48-50} and high-field asymmetric waveform ion mobility spectrometry (FAIMS)^{43,51,52} being two commonly utilized approaches. These enrichment approaches exploit the hydrophilic nature of glycans, in the case of ZIC-HILIC, or the shape of glycopeptide ions, in the case of FAIMS, to allow the separation of glycosylated peptides from non-glycosylated peptides improving the depth of glycoproteomic analysis. For both enriched and total proteome analyses, the resulting samples are analysed by LC-MS where in peptides/glycopeptides are subjected to MS based fragmentation, and the resulting spectra analysed using computational tools which match the resulting spectra to a protein database. While traditionally bottom-up proteomics using Data-Dependent Acquisition (DDA) has been used for bacterial proteome analysis⁵³⁻⁵⁵ and the analysis of glycopeptides^{38,39,41,43,48-52}, the stochastic nature of this method leads to missing datapoints across samples compromising quantitative analysis, especially within low abundant proteins⁵⁶. To address these issues Data-Independent Acquisition (DIA) has emerged as an alternative to DDA which leads to a dramatic reduction in missing datapoints across studies while enabling deep proteome analysis^{57,58}. Within DIA the highly reproducible sampling across the LC gradient enables the reproducible selection and fragmentation of peptides even allowing the detection of peptides not observable on the MS1 level^{59,60}, with recent improvements in DIA informatics broadening the accessibility of this approach^{57,61-63}. Combined these advancements now allow DIA to provide deep bottom-up proteomic coverage and track even challenging protein classes such as microproteins (<100 amino acid in length)⁶⁴ as well as low abundant proteins^{59,60}.”

5. Figure 1A is somewhat obsolete as the same information can be easily deduced from Figure 1B

The reviewer raises a fair point, in the interest of simplifying the figure we have now removed Figure 1A and updated the results section accordingly (see updated figure below).

6. A minor remark : the glycan marker ions in figure 1E are in the same color as c ions. This should be avoided.

We thank the reviewer for highlighting this oversight, we have now modified the glycan ion colours in figure 1D (formerly Figure 1E):

Updated Figure 1.

7. Figure 4A is in its present form not easy to read. In particular the grayscale used in the bottom part where correlation lines are drawn is not visible in a print on this scale.

We thank the reviewer for highlighting this oversight, we have now modified the shading and size of the points within the upset plot to improve readability.

Updated Figure 4.

8. With respect to Figure 4D, the enrichment analysis is done on the deltaFliF strain. While there are certainly similarities between this strain and the deltapglL strain, as shown from the heat map in figure 4B, there are some pertinent differences. It deserves to be demonstrated that the same enrichment for flagellum and chemotaxis proteins to be affected by FliF mutations are also true in the glycosylation mutant. At first sight, there seem to be some discrepancies in fold changes specifically of proteins involved in flagellin formation that can be easily linked to the fliF gene deletion. But the impact of glycosylation machinery deletion outside the glycoproteome is not as such evaluated at the GO level. This could hint to some non-obvious impact of loss of glycosylation as discussed on page 24.

The reviewer raises a fair point that differences in the flagellum and chemotaxis associated proteins altered between $\Delta fliF$ and $\Delta pglL$ may be occurring which are obscured by assessing this data at the GO term levels. It is true that just because the same functional groups are impacted across conditions it should not be assumed the same proteins are being impacted and efforts should be made to dissect what is occurring here. To provide a more granular assessment of these changes we have now generated heatmaps of all proteins associated with the GO terms “Chemotaxis [GO:0006935]” and “Bacterial-type flagellum-dependent cell motility [GO:0071973]” and provided this data as part of Supplementary Figure 19.

Examination of the proteins altered within these GO terms demonstrates while some proteins are the same this overlap is incomplete. We agree that this point should be highlighted and have modified the result sections to state (Line 553, Page 21):

“Examination of the individual proteins associated with the GO terms Chemotaxis [GO:0006935] and Bacterial-type flagellum-dependent cell motility [GO:0071973] supports similar changes between $\Delta pglL$ and $\Delta fliF$ compared to WT yet that the overlap in altered proteins is only partial (Supplementary Figure 19C and D).”

Updated Supplementary Figure 19

Editorial changes

1) Please insert the legends for Supplementary Tables in the first row of the Excel files (ie Table S1: [insert title]).

Titles have been added to the supplementary tables.

2) There are currently more than 10 supplemental material files uploaded. The total number of supplemental material files is limited to 10. Please either reduce the number of supplemental material files you have uploaded or combine them to meet the 10-file limitation. If you have additional items past the 10 permitted, we typically suggest uploading them to an external repository and citing them as necessary throughout the manuscript text file. Should you elect to take this route, please note we do not publish legends for supplemental materials that are not hosted by ASM. Therefore, any supplemental items that have not been uploaded to our submission system should not be included with the supplemental legend listed on the manuscript text file. Any material that is uploaded to an external repository must include the legend for said material on the repository host site. Also, please note you must include the external repository link

parenthetically in your manuscript anytime the supplemental material that is hosted there is being cited. For example, "(see Fig. S1 at [insert Figshare link here])."

The number of supplementary material files has been reduced to 10 now corresponding to Supplementary Material document and Supplementary Table 1 to 9. As instructed, the two Supplementary Data pdf files containing the annotated glycopeptide spectra have been moved to Figshare [please see <https://figshare.com/s/6dca9c47ec3b0cf2d586> and the currently private DOI: 10.6084/m9.figshare.25492774]. Throughout the document we have now highlighted the DOI associated with this Figshare link at:

Page 14, line 355

Page 15, line 361

Page 18, line 443

Page 19, line 461

Page 19, line 468

Page 25, line 611

3) Please remove the Supplementary Table 1-9 legends from lines 692-799 of your manuscript file. Please move those legends to the beginning of your Supplementary Material file, before Supplementary Material Tables 1-4.

The Supplementary Table Legends have been moved to the Supplementary Material document placed before Supplementary Tables 10-13.

4) Please change the names of the Supplementary Material Tables 1-4 so that they are Supplementary Tables 10-13.

Supplementary Material Tables 1-4 have been updated to be labeled Supplementary Tables 10-13 within the Supplementary Material document.

5) Please go through your manuscript and make sure that any citations to Supplementary Material Tables 1-4 are changed to reflect the change in name to Supplementary Tables 10-13.

The main document has been updated changing Supplementary Material Tables 1-4 to Supplementary Tables 10-13 at:

Page 8 Line 154

Page 9 Line 186

Page 9 Line 206

Page 10 Line 215

Page 18 Line 442

Re: Spectrum00346-24R1 (Glycoproteomic analysis of Burkholderia cenocepacia reveals the flagella apparatus proteins FliF and MotB are glycosylated yet glycosylation is dispensable for motility.)

Dear Prof. Nichollas E Scott:

Thank you for sending a revised version of your manuscript and addressing all reviewers' concerns. Your manuscript is almost ready to be accepted. However, I note a few points that I believe the reviewers missed and can further improve your work.

1. When reading the manuscript, I went through Figures 5 and 6, comparing the data, but the way the panels are presented is confusing. Consider presenting FliF and MotB motility in similar panels A and B, located in the same manner in both figures. This small change will greatly facilitate the comparison between the single and double mutants.
2. indicate the meaning of + and - in Figure 6 C and D (+ rhamnose? - rhamnose?)
3. The differential effect of inducing FliF or MotB in figures 5 and 6 is not described. It appears that even in noninducing conditions you have wild type levels of MotB and FliF (Figure 5). Upon induction it seems that overexpression of proteins occurs. The overexpression effect has an effect on the motility phenotype when the proteins are glycosylated. However, in the Δ pglL background this effect is abolished. Please consider adding a comment, possibly explaining the reasons at least in the discussion section.

Once these points are addressed, please return your submission (you may comment on my points in the cover letter) so that I can move your paper forward to acceptance.

Revision Guidelines

- Upload point-by-point responses to the issues raised by the editor in your cover letter.
- Upload a compare copy of the manuscript (without figures) as a "Marked-Up Manuscript" file.
- Upload a clean .DOC/.DOCX version of the revised manuscript and remove the previous version.
- Each figure must be uploaded as a separate, editable, high-resolution file (TIFF or EPS preferred), and any multipanel figures must be assembled into one file.
- Any supplemental material intended for posting by ASM should be uploaded with their legends separate from the main manuscript. You can combine all supplemental material into one file (preferred) or split it into a maximum of 10 files with all associated legends included.

Sincerely,

Silvia Cardona
Editor
Microbiology Spectrum

Response to Reviewers

Response to reviewers are denoted in red

Reviewer comments

Reviewer: 1

General comments

The manuscript is well written and well presented. The relevance and approach used are robust and the methods are provided with sufficient detail and statistical analysis of proteomics and glycoproteomics datasets is robust. An excellent manuscript with conclusions well supported by the results and figures.

We thank the reviewer for their positive and supportive comments on the manuscript.

Specific comments

Line 49 - define CF

We thank the reviewer for highlighting this oversight and have replaced CF with Cystic Fibrosis within the Importance statement.

Line 124 - the sentence is a little awkward. Consider revising.

We thank the reviewer for their suggestion, we have altered the sentence (Page 6, Line 137):

“Combined these advancements now allow DIA to provide deep proteomic coverage and the ability to precisely track protein classes traditionally challenging to monitor including microproteins (<100 amino acid in length)⁵⁷ as well as low abundant proteins^{52,53}.”

to

“Combined these advancements now allow DIA to provide deep proteomic coverage and track even challenging protein classes such as microproteins (<100 amino acid in length)⁵⁷ as well as low abundant proteins^{52,53}.”

Line 165 - write out E.coli in full for first introduction

We thank the reviewer for flagging this oversight, the full name, *Escherichia coli* has been added to line 165 (now line 177)

Line 514 (and throughout) - why are the three strains selected and generated (i.e., Δ motB, Δ fliF, and Δ BCAM0505) instead of all five of the glycoproteins. Justification is needed from the beginning.

We thank the reviewer for pointing out this oversight regarding the prioritization of glycoproteins for characterization. Out of the five glycoproteins [BCAL1086, BCAL2974, FliF

(BCAL0525), BCAM0505, and MotB (BCAL0127)] that exhibited changes in response to the loss of glycosylation, we have previously investigated the proteomic effects of the disruption of *BCAL1086* and *BCAL2974* in our prior (and more limited) proteomic study on how the absence of glycosylation affects *B. cenocepacia* K56-2 (Oppy *et al.* mSphere 2019). As such, within this study we focused on the glycoproteins newly identified to be sensitive to the loss of glycosylation from our DIA analysis [FliF (BCAL0525), BCAM0505, and MotB (BCAL0127)]. To clarify this point we have now added the following sentence to the results section (Page 21, Line 532):

“As our previous studies have demonstrated that the disruption of *BCAL1086* or *BCAL2974* does not drive the proteome changes observed in Δ *pglL*⁵⁵ we focused on the glycoproteins MotB, FliF, and BCAM0505.”

Line 515 - given the extensive information provided with the supplemental tables for the proteome assessment of the mutant vs. wild-type strains and the rather non-traditional approach to confirming a mutant (i.e., via proteomics) and not genetic approaches, I suggest including the fold-change or LFQ intensity values (to demonstrate absence or below the limit of detection for the protein within the mutant strain) within the main text. The values in the supplemental table appear to be imputed and so confirmation that the gene (and protein) are absent by LFQ is needed. Notably, the imputation is well represented within Supp. Fig. 13.

Our proteomic analysis of strains Δ *motB*, Δ *fliF*, and Δ BCAM0505 support that these three proteins are undetectable within mutants compared to wild type which is the key point highlighted in Supplementary Figures 13B, 14B and 15B yet we appreciate that this point has not clearly been made. To improve data transparency, we have now merged Supplementary tables 6, 7 and 8, for now known as Supplementary table 5, including the non-imputed data as well as updated the text of the main document to state (Page 20, Line 534):

“To assess if the loss of *motB*, *fliF*, and BCAM0505 contributes to the proteomic impact seen in the absence of glycosylation, we generated Δ *motB*, Δ *fliF*, and Δ BCAM0505 strains, and using DDA proteomic analysis confirmed the loss of these three proteins within mutants (Supplementary figures 13 to 15, Supplementary tables 5).”

It also should be noted that genetic approaches to confirm all mutants were undertaken, to make this point clearer we have now updated Page 10, Line 214 to 216 to state:

“Mutagenesis was confirmed using screening oligonucleotides which bind outside the region of recombination (Supplementary Document Table 3) using GoTaq DNA polymerase-based PCR screening supplemented with 10% DMSO.”

Line 555 - why confirm mutants and restoration by proteomics and not WGS to ensure that additional genetic disruptions were not made that may influence the changes in proteome remodeling upon comparison to wild-type?

We appreciate our use of proteomics to confirm mutants is unorthodox and admit our use of this approach is driven by our access to this technology. Within the lab after confirmation of mutagenesis using PCR based approaches our first pass to validate mutants, especially strains for use in large studies (such as multi-strain comparisons) is to undertake proteomics of independent mutants compared to WT strains. This approach does not replace our molecular (PCR based) approaches of confirming mutations but is intended to provide further and orthogonal confirmation of deletions. It should be noted with the increasing accessibility of WGS we are beginning to integrate this into our strain validation workflows, yet these strains have not been subjected to WGS. Additionally, complementation was observed to restore phenotypes, further supporting that the phenotypes assigned to our mutants are the result of the disruption in question.

Fig. 2 D-E - the labels on the red dots are too small to read - value in including them?

The reviewer raises a fair point, while we initially provided the individual measurements as red dots for data transparency, we agree that due to the complexity of this figure this does make these data points extremely small. As such we have updated the figure to use error bars in place of individual data points:

Updated Figure 2.

Reviewer #2 (Comments for the Author):

Glycosylation is quite recently recognized as a common post-translational modification in bacteria. Most studies were initiated in *Campylobacter jejuni*, but today it is understood

that this modification is widespread. Burkholderia belong to the species that are most studied in this context, but the exact size of the glycoproteome and its function is not well understood.

In this work, Lewis et al. used state-of-the-art mass spectrometric technology to perform a comprehensive analysis of the glycoproteome of species from the Burkholderia cenocepacia complex, known as an opportunistic pathogens to infect people with cystic fibrosis and related immunocompromised individuals. Not only do they provide data about the in depth identification of glycoproteins, they also used knock outs in the glycosylation machinery to investigate the impact of glycosylation on those proteins' abundance. Strikingly, while the global proteome undergoes a somewhat unexpected large change in abundance, proteins that are glycosylated do not change in abundance, hinting that glycosylation does not have an impact on their stability. Exceptions herein are the previously known glycoproteins MotB and FliF involved in motility. Additional studies on knock outs for the corresponding genes or specific mutation of their glycosite indicate are studies, revealing that in contrast what was believed before, that glycosylation of MotB and FliF has no impact on motility. While the latter may seem a negative result, together with the unprecedented powerful and comprehensive analysis of the overall Burkholderia cenocepacia glycoproteome and report of the impact of loss of glycosylation in general, it sheds new lights on the function of bacterial glycosylation.

While the overall setup, methodology and the quality of data and reporting of results are excellent, there are some items that should be better clarified or improved, as well as some additional information might be highly useful. Below these are listed in chronological order of appearance in the text.

We thank the reviewer for their kind and supportive comments on the manuscript.

1. At some instances, the mass spectrometric data are searched against the J2315 database, whereas in other analyses both K56-2 and J2315 proteome databases are searched. Since the former contains more proteins, why has the glycopeptide identification not been performed in this database. This reviewer is aware that Uniprot merged most K56-2 entries in the J2315 database, but still it is known that the K56-2 strain is genetically somewhat different. It is also not clear whether the two databases were merged or whether they were individually searched for the DIA an DDA quantitative analyses. This should be clarified in the method section.

The review raises an excellent point on the databases used for analysis.

For glycopeptide focused analysis the J2315 database was used in line with our previous glycoproteomic studies (Ahmad Izaham *et al.* Mol Cell Proteomics. 2020; Ahmad Izaham *et al.* J Proteome Res. 2021) yet we appreciate that this may lead to the loss of some glycopeptides due to variations in the peptide sequences between J2315 and K56-2. While our previous analysis of site conservation suggests that at the peptide level there is very little alteration between strains within the region flanking glycosylation sites (>95% identity across 294 *B. cenocepacia* genomes within 69 sites, see Hayes *et al.* Commun Biol. 2021; 4: 1045.) there may be cases where glycopeptides are missed due to sequence variation. This

does not undermine the novel glycopeptides identified but may mean additional glycoproteins could be identified if the K56-2 database was used to assess these glycoproteomic datasets. To ensure readers are aware of this caveat we have now added the following sentence (Page 18, Line 453):

“Glycopeptides were identified against the J2315 proteome with our previous work highlighting the high sequence identity within the surrounding amino acids of glycosylation sites within *B. cenocepacia*³⁸, however this approach is insensitive to the detection of glycopeptides unique to K56-2.”

For DIA and DDA quantitative analyses both the J2315 and K56-2 databases were searched together, effectively merging these databases. By searching these databases together information on the identified proteins/peptides are assigned to both J2315 and K56-2 assignments which streamlines the matching of proteins to known J2315 accessions. This point has now been highlighted in (Page 15, line 363):

“Data files were searched against the *B. cenocepacia* K56-2 proteome (Uniprot accession: UP000011196, 7467 proteins, downloaded October 13th 2020)⁹⁴ and *B. cenocepacia* strain J2315 (Uniprot accession: UP000001035)⁹⁰, effectively merging these proteomes enabling the matching of proteins to both J2315 and K56-2 accessions”

2. It should be noted when the Uniprot databases were uploaded. Recently, the annotations of the Burkholderia J2315 proteins improved, with more previously hypothetical/putative proteins being annotated. This particularly refers to the statements at page 17 saying that half of the discovered glycoproteins were unknown. Did the authors make an effort to manually annotated these proteins. More specifically, concerning the lack of GO ontology terms, it could have been of interest to investigate the presence of signal peptides or membrane anchors that could point to cellular location as the glycosylation machinery is supposed to be periplasmic. Are all identified proteins supposed to be periplasmic/membrane oriented or secreted ?

We thank the reviewer for the suggestion to clearly outline when the Uniprot databases used in this study were downloaded, we have now provided these dates adding these details to Page 14, line 346:

“*B. cenocepacia* reference proteome J2315 Uniprot accession: UP000001035, 6,993 proteins, downloaded July 24th 2020”

And to Page 15, line 365:

“*B. cenocepacia* K56-2 proteome (Uniprot accession: UP000011196, 7467 proteins, downloaded October 13th 2020”

Regarding our statement on Page 17 we apologize to the reviewer for our poor choice of wording. The glycoproteins highlighted here as unknown do not correspond to open reading frames which were previously unassigned within the J2315 proteome but to known proteins

that were unrecognized as glycosylated. To highlight this point, we have modified the sentence on (Page 18, Line 467) from:

“revealing 12 of the 54 glycoproteins identified within this study correspond to previously unknown *B. cenocepacia* glycoproteins (Figure 1D, Supplementary Figure 3).”

to

“revealing 12 of the 54 glycoproteins identified within this study correspond to previously unrecognized *B. cenocepacia* glycoproteins (Figure 1C, Supplementary Figure 3).”

Based on the periplasmic location of the oligosaccharyltransferase all protein subjected to glycosylation are exposed to the periplasmic space. To help summarise the details on the predicted number of transmembrane domains, the presence of signal peptides as well as the GO terms assigned to proteins of the known glycoproteome, we have now updated Supplementary table 2 to contain these details. Consistent with the expectation that proteins are secreted we observe most glycoproteins contain transmembrane domains, a Signal tag or are assigned as membrane associated proteins based on GO terms. We have now highlighted that these details are provided in Supplementary Table 2, page 27 line 704:

“Supplementary Table 2. Glycoproteins identified across *B. cenocepacia*. Curated list of all known glycoproteins within *B. cenocepacia* including confirmed glycosylation sites, residue of the assigned glycosylation site and if the glycoproteins have been previously identified within previous studies. For each glycoprotein the location and number of predicted transmembrane domains, location of the Signal tag and GO terms associated with each protein are provided derived from the *B. cenocepacia* J2315 proteome (UP000001035).”

3. The method section should be checked for completeness, in particularly for the database searching. For example, supplemental data refer to Byonic searches, but this is not included in the methodology.

We apologise to the reviewer for the confusion on the origins of the Byonic glycopeptide assignments. To ensure the glycopeptides/glycoproteins identified within this study were truly unique we curated a list of all known glycoproteins within *B. cenocepacia* identified across previous studies (Supplementary table 2). During this curation to ensure only true glycoproteins were included in this final list we manually validated all unique assignments which had not been previously manually validated in the studies Ahmad Izaham *et al.* Mol Cell Proteomics. 2020 and Ahmad Izaham *et al.* J Proteome res. 2021 where glycopeptides were identified using Byonic. To clearly highlight the origin of these glycopeptide assignments we have added the following sentences to the method section (page 14, line 354):

“To assess the uniqueness of identified glycoproteins within this study we curated a list of all previously known *B. cenocepacia* glycoproteins^{38,39,41,43}. Unique glycopeptides identified using the glycopeptide identification tool Byonic (Protein Metrics Inc.⁹³) yet not previously manually confirmed from studies^{39,43} were annotated to ensure correctness resulting in a

total of 129 unique glycoproteins across all studies^{38,39,41,43} (Supplementary Table 2) with unique glycopeptides from these previous Byonic searches provided within Supplementary Data 2.”

4. In this respect, the authors are invited to provide some extra information on the different ms methods and data analysis approaches that are used. This manuscript is submitted to a more general microbial journal and the approaches and result reports require high level knowledge of proteomics methodologies generally not known to microbiologists.

We thank the reviewer for this suggestion, and we have now expanded the introduction to provide more information and background to readers on the MS methods used particularly glycoproteomic analysis, Page 5, Line 114:

“The analysis of bacterial proteomes using liquid chromatography–mass spectrometry (LC-MS) has emerged as the quintessential approach for the quantitation of microbial proteomes and post-translational modifications^{44,45}. Within bottom-up proteomics, the most widely implemented form of proteomics, proteins are first digested into peptides prior to analysis⁴⁶. Once digested, peptides can be analysed directly or subjected to enrichment approaches to enhance the detection of specific peptides of interest, such as glycopeptides in the case of glycoproteomics⁴⁷. In glycoproteomics a range of enrichment approaches can be used to isolate glycopeptides⁴⁷ with Zwitterionic hydrophilic interaction liquid chromatography (ZIC-HILIC)^{38,39,41,48-50} and high-field asymmetric waveform ion mobility spectrometry (FAIMS)^{43,51,52} being two commonly utilized approaches. These enrichment approaches exploit the hydrophilic nature of glycans, in the case of ZIC-HILIC, or the shape of glycopeptide ions, in the case of FAIMS, to allow the separation of glycosylated peptides from non-glycosylated peptides improving the depth of glycoproteomic analysis. For both enriched and total proteome analyses, the resulting samples are analysed by LC-MS where in peptides/glycopeptides are subjected to MS based fragmentation, and the resulting spectra analysed using computational tools which match the resulting spectra to a protein database. While traditionally bottom-up proteomics using Data-Dependent Acquisition (DDA) has been used for bacterial proteome analysis⁵³⁻⁵⁵ and the analysis of glycopeptides^{38,39,41,43,48-52}, the stochastic nature of this method leads to missing datapoints across samples compromising quantitative analysis, especially within low abundant proteins⁵⁶. To address these issues Data-Independent Acquisition (DIA) has emerged as an alternative to DDA which leads to a dramatic reduction in missing datapoints across studies while enabling deep proteome analysis^{57,58}. Within DIA the highly reproducible sampling across the LC gradient enables the reproducible selection and fragmentation of peptides even allowing the detection of peptides not observable on the MS1 level^{59,60}, with recent improvements in DIA informatics broadening the accessibility of this approach^{57,61-63}. Combined these advancements now allow DIA to provide deep bottom-up proteomic coverage and track even challenging protein classes such as microproteins (<100 amino acid in length)⁶⁴ as well as low abundant proteins^{59,60}.”

5. Figure 1A is somewhat obsolete as the same information can be easily deduced from Figure 1B

The reviewer raises a fair point, in the interest of simplifying the figure we have now removed Figure 1A and updated the results section accordingly (see updated figure below).

6. A minor remark : the glycan marker ions in figure 1E are in the same color as c ions. This should be avoided.

We thank the reviewer for highlighting this oversight, we have now modified the glycan ion colours in figure 1D (formerly Figure 1E):

Updated Figure 1.

7. Figure 4A is in its present form not easy to read. In particular the grayscale used in the bottom part where correlation lines are drawn is not visible in a print on this scale.

We thank the reviewer for highlighting this oversight, we have now modified the shading and size of the points within the upset plot to improve readability.

Updated Figure 4.

8. With respect to Figure 4D, the enrichment analysis is done on the deltaFliF strain. While there are certainly similarities between this strain and the deltapglL strain, as shown from the heat map in figure 4B, there are some pertinent differences. It deserves to be demonstrated that the same enrichment for flagellum and chemotaxis proteins to be affected by FliF mutations are also true in the glycosylation mutant. At first sight, there seem to be some discrepancies in fold changes specifically of proteins involved in flagellin formation that can be easily linked to the fliF gene deletion. But the impact of glycosylation machinery deletion outside the glycoproteome is not as such evaluated at the GO level. This could hint to some non-obvious impact of loss of glycosylation as discussed on page 24.

The reviewer raises a fair point that differences in the flagellum and chemotaxis associated proteins altered between ΔfliF and ΔpglL may be occurring which are obscured by assessing this data at the GO term levels. It is true that just because the same functional groups are impacted across conditions it should not be assumed the same proteins are being impacted and efforts should be made to dissect what is occurring here. To provide a more granular assessment of these changes we have now generated heatmaps of all proteins associated with the GO terms “Chemotaxis [GO:0006935]” and “Bacterial-type flagellum-dependent cell motility [GO:0071973]” and provided this data as part of Supplementary Figure 19.

Examination of the proteins altered within these GO terms demonstrates while some proteins are the same this overlap is incomplete. We agree that this point should be highlighted and have modified the result sections to state (Line 553, Page 21):

“Examination of the individual proteins associated with the GO terms Chemotaxis [GO:0006935] and Bacterial-type flagellum-dependent cell motility [GO:0071973] supports similar changes between $\Delta pglL$ and $\Delta fliF$ compared to WT yet that the overlap in altered proteins is only partial (Supplementary Figure 19C and D).”

Updated Supplementary Figure 19

Editorial changes

1) Please insert the legends for Supplementary Tables in the first row of the Excel files (ie Table S1: [insert title]).

Titles have been added to the supplementary tables.

2) There are currently more than 10 supplemental material files uploaded. The total number of supplemental material files is limited to 10. Please either reduce the number of supplemental material files you have uploaded or combine them to meet the 10-file limitation. If you have additional items past the 10 permitted, we typically suggest uploading them to an external repository and citing them as necessary throughout the manuscript text file. Should you elect to take this route, please note we do not publish legends for supplemental materials that are not hosted by ASM. Therefore, any supplemental items that have not been uploaded to our submission system should not be included with the supplemental legend listed on the manuscript text file. Any material that is uploaded to an external repository must include the legend for said material on the repository host site. Also, please note you must include the external repository link

parenthetically in your manuscript anytime the supplemental material that is hosted there is being cited. For example, "(see Fig. S1 at [insert Figshare link here])."

The number of supplementary material files has been reduced to 10 now corresponding to Supplementary Material document and Supplementary Table 1 to 9. As instructed, the two Supplementary Data pdf files containing the annotated glycopeptide spectra have been moved to Figshare [please see <https://figshare.com/s/6dca9c47ec3b0cf2d586> and the currently private DOI: 10.6084/m9.figshare.25492774]. Throughout the document we have now highlighted the DOI associated with this Figshare link at:

Page 14, line 355

Page 15, line 361

Page 18, line 443

Page 19, line 461

Page 19, line 468

Page 25, line 611

3) Please remove the Supplementary Table 1-9 legends from lines 692-799 of your manuscript file. Please move those legends to the beginning of your Supplementary Material file, before Supplementary Material Tables 1-4.

The Supplementary Table Legends have been moved to the Supplementary Material document placed before Supplementary Tables 10-13.

4) Please change the names of the Supplementary Material Tables 1-4 so that they are Supplementary Tables 10-13.

Supplementary Material Tables 1-4 have been updated to be labeled Supplementary Tables 10-13 within the Supplementary Material document.

5) Please go through your manuscript and make sure that any citations to Supplementary Material Tables 1-4 are changed to reflect the change in name to Supplementary Tables 10-13.

The main document has been updated changing Supplementary Material Tables 1-4 to Supplementary Tables 10-13 at:

Page 8 Line 154

Page 9 Line 186

Page 9 Line 206

Page 10 Line 215

Page 18 Line 442

Re: Spectrum00346-24R2 (Glycoproteomic analysis of Burkholderia cenocepacia reveals the flagella apparatus proteins FlIF and MotB are glycosylated yet glycosylation is dispensable for motility.)

Dear Prof. Nichollas E Scott:

Your manuscript has been accepted, and I am forwarding it to the ASM production staff for publication. Your paper will first be checked to make sure all elements meet the technical requirements. ASM staff will contact you if anything needs to be revised before copyediting and production can begin. Otherwise, you will be notified when your proofs are ready to be viewed.

Sincerely,
Silvia Cardona
Editor
Microbiology Spectrum